# DOCREWARD: A DOCUMENT REWARD MODEL FOR STRUCTURING AND STYLIZING

## ABSTRACT

Recent advances in agentic workflows have enabled the automation of tasks such as professional document generation. However, they primarily focus on textual quality, neglecting visual structure and style, which are crucial for readability and engagement. This gap arises mainly from the absence of suitable reward models to guide agentic workflows toward producing documents with stronger structural and stylistic quality. To address this, we propose DOCREWARD, a *Document Reward Model* that evaluates documents based on their structure and style. We construct a multi-domain dataset DOCPAIR of 117K paired documents, covering 32 domains and 267 document types, each including a high- and low-professionalism document with identical content but different structure and style. This enables the model to evaluate professionalism comprehensively, and in a textual-quality-agnostic way. DOCREWARD is trained using the Bradley-Terry loss to score documents, penalizing predictions that contradict the annotated ranking. To assess the performance of reward models, we create a test dataset containing document bundles ranked by well-educated human evaluators. Notably, DOCREWARD outperforms GPT-4o and GPT-5 in accuracy by **30.6** and **19.4** percentage points, respectively, demonstrating its superiority over baselines. In extrinsic evaluations, both re-ranking and RL experiments demonstrating its utility in guiding generation agents toward producing human-preferred documents.

## 1 INTRODUCTION

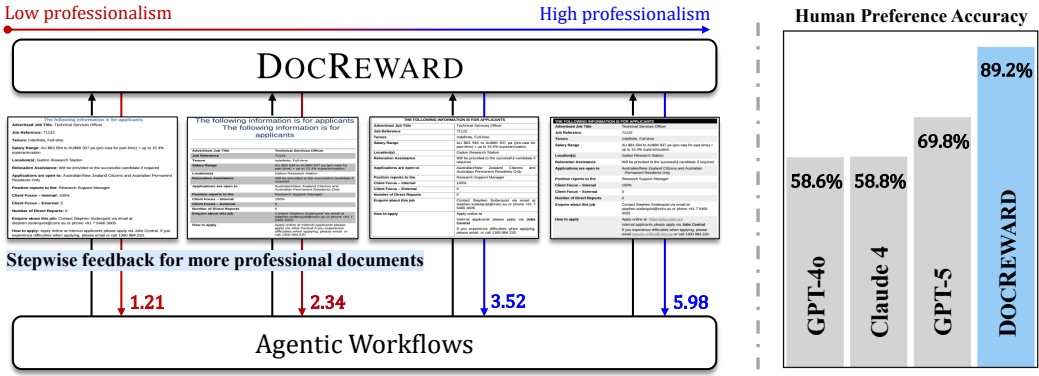

Figure 1: DOCREWARD automatically assesses document professionalism according to their structure and style, assisting existing agentic workflows for more professional document generation (left). It outperforms GPT-5 by 19.4% in human preference accuracy (right).

Recent advances in agentic workflows have automated many complex tasks, such as code generation (Peng et al., 2023; Cherny & Anthropic, 2025; Hong et al., 2024), image generation (comfyanonymous, 2025), visual understanding (Zheng et al., 2025; Marsili et al., 2025), math reasoning (Yan et al., 2025), and travel planning (Xie et al., 2024). A key focus of agentic workflows is the production of professional documents, including works like deep research (OpenAI, 2025a; Liang et al., 2025; Qwen, 2025) and technical documentation generation (Dvivedi et al., 2024). However,

existing research about professional document generation primarily focuses on improving textual quality, neglecting the importance of visual structure and style, both of which play crucial roles in shaping document professionalism. A well-organized structure helps the reader navigate the material smoothly, while a consistent style makes the content more readable and engaging. Together, these aspects help convey information more clearly and effectively. The neglect of structure and style mainly stems from the lack of suitable reward models, which are capable of guiding agentic workflows to produce documents with more professional structure and style.

To address this, we propose DOCREWARD, a ***Doc**ument **Reward** Model*, specialized in assessing document professionalism in structure and style, as shown in Figure 1. However, building a reward model capable of providing a robust evaluation of visual structure and style is non-trivial, as it requires both *comprehensiveness* and *textual-quality-agnosticism*. Specifically, comprehensiveness refers to the ability to evaluate documents across diverse types, qualities, structures, and styles, while textual-quality-agnosticism, in this context, means that the model does not evaluate the inherent quality of the textual content itself, but instead assesses how well the structure and style of a document stand out, given the fixed content.

To achieve both comprehensiveness and textual-quality-agnosticism, we construct a multi-domain dataset, DOCPAIR, consisting of 117K paired documents, covering 32 domains and 267 document types, with each pair consisting of a high-professionalism sample and its low-professionalism counterpart. The paired documents share identical content but differ in structure and style. The construction of DOCPAIR consists of three phases: 1) *Curating High-Quality Professional Documents.* We curate a set of high-quality documents with strong professionalism in structure and style, from various domains (*e.g.*, government, education, science, etc.) 2) *Expanding Source Documents via Agents.* Next, we extract both the textual content and the rendered pages of the source documents. Subsequently, multiple generation agents are prompted to produce a new document that preserves the textual content of the original and adheres to appropriate structure and style. 3) *Ranking Documents.* When comparing a source document with its generated counterparts, the original human-authored version is always preferred. In other cases, we use the original professional document as a reference and employ GPT-5 (OpenAI, 2025b) to rank document bundles by their structural and stylistic professionalism.

Based on the constructed dataset, we train DOCREWARD to take rendered document pages as inputs and output a score reflecting the document's professionalism in structure and style. The predicted scores of paired documents are optimized using the Bradley-Terry loss (Bradley & Terry, 1952; Ouyang et al., 2022), which penalizes violations of the annotated order.

To demonstrate the superiority and utility of DOCREWARD, we perform both intrinsic and extrinsic evaluations. For intrinsic evaluation, we create a test set of 473 human-annotated pairs across multiple document domains. Each pair is ranked by expert human annotators according to the professionalism of the paired documents' structure and style. Notably, as shown in Figure 1 (right), DOCREWARD outperforms GPT-4o (Hurst et al., 2024) and GPT-5 (OpenAI, 2025b) by 30.6 and 19.4 percentage points, respectively, in accuracy on the test set, demonstrating its superiority over existing approaches. For extrinsic evaluation, we evaluate DOCREWARD through two complementary experiments. 1) DOCREWARD is used as a re-ranking model for improving agentic workflow without changing the agent itself. A human evaluation shows that DOCREWARD as a reward model achieves a significantly higher win rate of 60.8%, compared to GPT-5's 37.7%. 2) We further demonstrate the utility of DOCREWARD as the reward model for reinforcement-learning of both open- and closed-source agentic workflows. This integration improves the document generation performance of Qwen2.5-Coder and GPT-4o in terms of structure and style. To conclude, the above experiments demonstrate that DOCREWARD can guide generation agents to produce human-preferred documents, making it a valuable tool to improve document generation.

The contributions of this paper are summarized as follows:

- We propose DOCREWARD, a ***Doc**ument **Reward** Model* specialized in assessing document professionalism in terms of structure and style.

- To equip DOCREWARD with comprehensiveness and textual-quality-agnosticism, we construct a multi-domain dataset DOCPAIR, consisting of 117K paired documents across 32 domains and 267 document types. This enables the model to evaluate professionalism in structure and style comprehensively and independently of inherent textual content quality.

- Comprehensive experiments demonstrate that DOCREWARD not only surpasses GPT-4o and GPT-5 in evaluating document professionalism in terms of structure and style, but also serves effectively as a reward model for RL, improving the document generation performance of both open-source and closed-source agentic workflows.

## 2 TASK FORMULATION

A document's professionalism is determined by its textual content, structure, and style. Although large language models excel at evaluating textual quality, they are limited in assessing structure and style. To bridge this gap, we develop reward models tailored to these dimensions to advance agentic workflows in producing documents with more professional structure and style. In this section, we formulate the task and provide a clear definition of its objectives.

Let $\{D_i\}_{i=1}^N$ denote a set of $N$ documents, where each document $D_i$ consists of textual content $D_{\text{text},i}$ and rendered images $D_{\text{img},i}$. The document reward model $\mathcal{R}_\theta$ assigns scores to documents that share the same textual content, such that the scores reflect their structural and stylistic quality. This process is formalized as follows:

$$\max_\theta \ \text{Sim}\big(\pi^*, \text{Argsort}(\mathcal{R}_\theta(D_{img,1}), \mathcal{R}_\theta(D_{img,2}), \ldots, \mathcal{R}_\theta(D_{img,N}))\big) \tag{1}$$

$$\text{s.t. } D_{\text{text},i} = D_{\text{text},j}, \forall i, j,$$

where "Sim" is a predefined similarity function that measures the agreement between the true and predicted quality orders. "Argsort" returns the indices of documents sorted by their predicted scores. $\pi^*$ denotes the true indices reflecting the relative ranking of the documents in terms of structure and style.

In this paper, document professionalism in structure and style is defined as follows:

- *Structure:* Proper use of white space, appropriate margins, clear section breaks, well-structured text alignment, adequate paragraph spacing, proper indentation, inclusion of page headers and footers, and logical, coherent organization of content.

- *Style:* Appropriate font choices (type, size, color, readability), clear heading styles, effective use of emphasis (bold, italics), bullet points, numbering, and consistent formatting.

By optimizing $\mathcal{R}_\theta$ based on these factors, we obtain a reward model capable of assessing the structural and stylistic professionalism in a comprehensive and textual-quality-agnostic way.

## 3 DOCREWARD

We propose DOCREWARD, a reward model specializing in assessing the structural and stylistic professionalism of documents. DOCREWARD is trained on DOCPAIR, a diverse dataset of 117K document pairs (Section 3.1), and is optimized with a preference-based objective for structural and stylistic assessment (Section 3.2). The following sections detail the data construction pipeline and model design.

### 3.1 DATA CONSTRUCTION

As shown in Figure 2, we first collect a set of high-quality real-world source documents. The source documents are then expanded by multiple generation agents, and the resulting documents are grouped by shared textual content. Finally, each group of documents is annotated with a ranking $\pi^*$ in terms of structure and style quality. The overall process results in DOCPAIR, a dataset comprising 117K document pairs, covering 32 domains and 267 document types. The construction procedure is detailed step by step below:

**Curating High-Quality Professional Documents.** As illustrated in Figure 2 (top), we first curate a corpus of human-authored Microsoft Word documents that spans both highly formal institutional writing and everyday professional communication. We draw on two complementary sources:

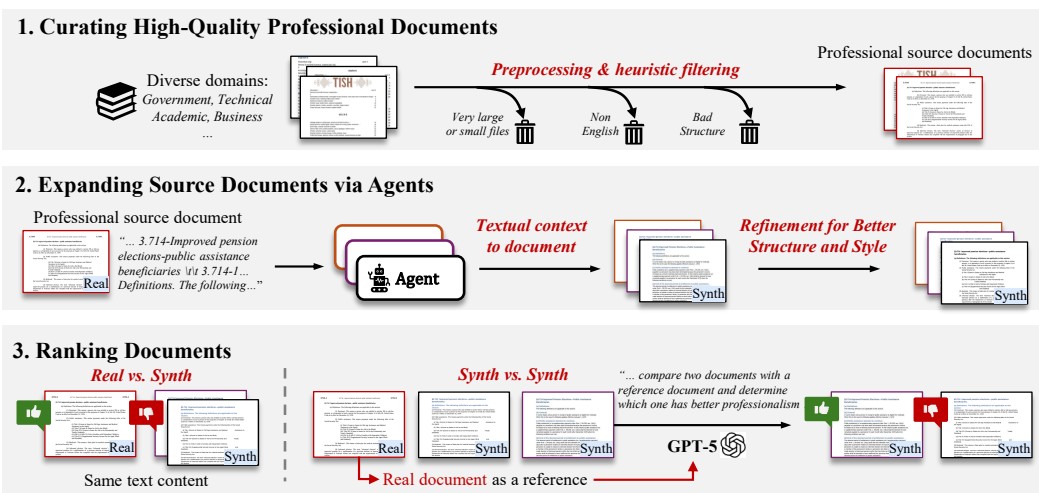

Figure 2: The data construction pipeline for DOCREWARD.

- *Government and institutional corpora:* GovDocs1 (Garfinkel et al., 2009) and NapierOne (Davies et al., 2022). GovDocs1 is a publicly available collection compiled from U.S. government (`.gov`) websites, including policy reports, administrative forms, statistical reports, public guidance, and meeting minutes, etc. NapierOne is a modern, comprehensive document dataset sourced from a wide range of public institutional materials and common office documents. These corpora provide authoritative, consistently professional exemplars of document structure and style.

- *Web document corpus:* We also draw from a diverse set of documents discovered in the CommonCrawl repository[1]. This corpus captures a broad range of real-world professional documents from business, education, nonprofit, healthcare, and other sectors, such as proposals, syllabi, newsletters, technical manuals, and policy briefs. It substantially enhances structural and stylistic diversity across professional genres.

To ensure suitability for reward-model training, we apply a light-weight preprocessing and filtering pipeline before data construction. First, all files are converted to DOCX format to enable programmatic access and modification via PYTHON-DOCX[2]. Next, we discard extreme or malformed cases (exceeding 20 pages, files larger than 1 MB dominated by images, and files smaller than 10 KB with trivial content). To efficiently reduce residual noise, we employ GPT-5 as a rigorous automated heuristic to flag poor structure/style on a $[0, 10]$ scale; documents scoring above 8 are retained. A manual inspection of 200 randomly sampled retained documents confirms that this automated filter preserves high-quality professional samples.

Finally, we analyze the distribution of domains and document types to assess coverage. The filtered collection spans 32 domains (e.g., government, education, nonprofit, medical, scientific, legal, business, academic, technical) and over 267 document types (e.g., job descriptions, government forms, policy documents, meeting minutes, press releases, course syllabi). The top 10 domains and top 30 document types are shown in Figure 3 and Figure 4, respectively, demonstrating both breadth and diversity. These high-quality, professional documents form the foundation for constructing subsequent document bundles and comparison pairs.

**Expanding Source Documents via Agents.** As shown in Figure 2 (middle), to obtain documents with the same textual content but different structure and style, we construct two types of agents to synthesize new documents given the textual content (and rendered pages) of the source documents. To further increase the diversity of the synthesized documents, each agent can be empowered by different LLMs. The two proposed agents are detailed as follows:

---

[1] https://commoncrawl.org/
[2] https://python-docx.readthedocs.io/en/latest/

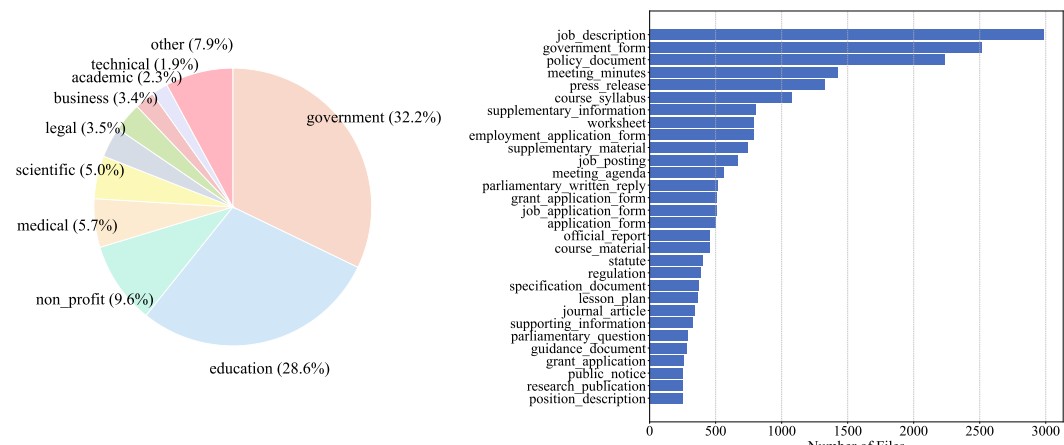

Figure 3: Top 10 Document Domain Distribution (Total: 32).

Figure 4: Top 30 Document Type Distribution.

- *Textual Content to Document.* The textual content is first extracted from the source documents, discarding all formatting, styling, and layout information. Then, advanced generation agents (*e.g.*, GPT-4o, OpenAI o1 (OpenAI, 2024), Claude Sonnet 4 (Anthropic, 2025), and GPT-5) are used to synthesize DOCX documents via PYTHON-DOCX. This process simulates the real-world task of generating professionally structured and styled documents from plain text.

- *Refinement for Better Structure and Style.* To further improve the structure and style of synthesized documents, we refine them by comparing with the original human-authored documents in terms of structure and style. The refinement process consists of two stages: 1) Generation agents are provided with the PYTHON-DOCX code, rendered pages, and structured textual representation of the synthesized document, along with the rendered pages of the original human-authored document, to generate a refinement plan. 2) Using this refinement plan, the agents modify the PYTHON-DOCX code to produce refined documents with better structure and style.

Since generation agents may omit textual content from the original documents, we remove any synthesized documents whose textual content deviates significantly from that of the original human-authored one. The remaining synthesized documents are then grouped with their originals to facilitate subsequent processing. For the details and prompts of this phase, please refer to Appendix A.3 and Appendix A.6.

**Ranking Documents.** As shown in Figure 2 (bottom), the collected documents within the same group share identical textual content and are organized into pairs. The annotation task is to assess the relative professionalism in terms of structure and style for each pair, which is carried out under the following two cases:

- *Real v.s. Synth.* If any sample in the pair is from the human-authored professional documents 3.1, it is directly designated as the preferred (winner).

- *Synth v.s. Synth.* When both samples in the pair are generated by agents, we prompt GPT-5 with a document triplet $\{D_{\text{real}}, D_{\text{synth1}}, D_{\text{synth2}}\}$, where the human-authored professional document $D_{\text{real}}$ is used as a reference to decide which synthetic sample is preferred. GPT-5 achieves an average accuracy of 92.5% on a human-annotated evaluation set consisting of 120 pairs in our preliminary test, demonstrating that the triple-wise annotation method is reliable and well-aligned with human judgment. The prompt is presented in Appendix A.6.

The two types of annotations are both guided by human-authored professional documents, and serve complementary purposes: "*Real vs. Synth*" pairs steer agentic workflows toward human-level document generation, while "*Synth vs. Synth*" pairs promote self-refinement. The data statistics of the constructed dataset, *i.e.*, DOCPAIR, are shown in Table 1.

| Domains | Doc. Types | Docs | Avg. Page | Doc. Pairs | | |
|---------|-----------|------|-----------|------------|------------------|-------------------|
| | | | | Total | Real *vs.* Synth | Synth *vs.* Synth |
| 32 | 267 | 69,137 | 3.2 | 117,108 | 36,664 | 80,444 |

Table 1: Data statistics of the constructed DOCPAIR.

## 3.2 MODEL STRUCTURE AND OPTIMIZATION

We adopt Qwen-2.5-VL (Bai et al., 2025) as the base model due to its advanced native multi-image input capabilities, which allow for a more comprehensive analysis of multi-page documents. An N-page document is converted into N images, which are then input into the model. A regression head is added to predict a scalar score on top of the output hidden states. More implementation details are presented in Appendix A.2.

We optimize DOCREWARD using the Bradley-Terry (BT) loss, which is specifically designed for learning from pairwise preferences. Specifically, let $D_{\text{img}}^w$ and $D_{\text{img}}^l$ be the rendered pages of the preferred (winner) and those of the less preferred (loser) in a paired comparison, respectively, then, the DOCREWARD (formatted as $\mathcal{R}_\theta$), takes in the rendered pages of each document and outputs scores, separately, which are supervised with the following objective:

$$\min_\theta - \log \sigma \big( \mathcal{R}_\theta(D_{\text{img}}^w) - \mathcal{R}_\theta(D_{\text{img}}^l) \big), \tag{2}$$

where $\sigma$ is the sigmoid function, defined as $\sigma(x) = \frac{1}{1+e^{-x}}$. This objective encourages the model to assign a higher score to the preferred document compared to the less preferred one.

## 4 EXPERIMENTS

We conduct experiments to evaluate the effectiveness of DOCREWARD in assessing both structural and stylistic professionalism of documents. This section includes evaluation dataset annotation, quantitative comparisons with strong baselines, extrinsic evaluation of document generation, and qualitative analyses.

### 4.1 EVALUATION DATASET COLLECTION AND HUMAN ANNOTATION

A subset of the curated documents in Section 3.1 is set aside as evaluation documents. To diversify the evaluation dataset, we consider the following six types of documents using the method described in Section 3.1. Four of them are obtained via the *Textual Content to Document* agent, which generates DOCX documents using different LLMs (*e.g.*, GPT-4o, OpenAI o1, Claude Sonnet 4, and GPT-5). One type comes from the *Refinement for Better Structure and Style* agent, where GPT-5 is employed to refine synthesized documents. The last type consists of the original human-authored documents. Together, these six types constitute the origins of samples in our evaluation dataset. For each set of documents sharing the same content but differing in structure and style, human experts meticulously rank their quality based on structure and style. To facilitate model evaluation, these ranked relationships are converted into a total of 473 comparison pairs, each consisting of two documents and a binary label indicating the preferred one. To ensure the quality of human annotation, two highly educated annotators annotate the same subset of documents; then, we evaluate annotation consistency among human annotators using Cohen's Kappa and observe a high agreement of 83.4. The detailed inter-annotator agreement results are presented in Table 5.

### 4.2 BASELINES AND EVALUATION SETTINGS

We evaluate our approach against several strong language models, including GPT-4o, Claude Sonnet 4, and GPT-5. Two evaluation settings are considered: *pairwise* and *pointwise*. In the pairwise setting, the model receives the rendered pages of two documents and is instructed to predict which document exhibits superior structure and style. In the pointwise setting, the model is provided with the rendered pages of a single document and assign a scalar score for structure and style without any reference document. The evaluation metric is accuracy, defined as the proportion of predictions that correctly match human annotations in the evaluation dataset.

| Model | Human Preference Accuracy (%) | | |
|---|---|---|---|
| | Synth vs. Synth | Real vs. Synth | Overall |
| **Pairwise Setting** | | | |
| Qwen2.5-VL-3B | 47.03 | 60.89 | 54.97 |
| Qwen2.5-VL-7B | 52.97 | 61.62 | 57.93 |
| GPT-4o (Hurst et al., 2024) | 58.91 | 66.43 | 63.22 |
| Claude Sonnet 4 (Anthropic, 2025) | 57.86 | 69.02 | 64.26 |
| GPT-5 (OpenAI, 2025b) | 64.78 | 72.32 | 69.1 |
| **Pointwise Setting** | | | |
| Qwen2.5-VL-3B | 36.63 | 33.58 | 34.88 |
| Qwen2.5-VL-7B | 41.58 | 57.93 | 50.95 |
| GPT-4o (Hurst et al., 2024) | 50.99 | 64.21 | 58.56 |
| Claude Sonnet 4 (Anthropic, 2025) | 48.02 | 66.79 | 58.77 |
| GPT-5 (OpenAI, 2025b) | 64.85 | 73.43 | 69.77 |
| DOCREWARD-3B (Ours) | 72.77 | 97.42 | 86.89 |
| DOCREWARD-7B (Ours) | **78.22** | **97.42** | **89.22** |

Table 2: Accuracy of Models on the proposed evaluation dataset. 'Real vs. Synth' represents evaluation pairs where a human-authored document is compared against a document generated by an agent. 'Synth vs. Synth' represents evaluation pairs where two agent-generated documents are compared.

| Reward Models | Win | Lose | Tie |
|---|---|---|---|
| Random | 24.6 | 66.2 | 9.2 |
| GPT-5 | 37.7 | 40.0 | 22.3 |
| DOCREWARD (Ours) | 60.8 | 16.9 | 22.3 |

Table 3: Extrinsic evaluation results. DOCREWARD shows utility for professional document generation.

### 4.3 RESULTS ON EVALUATION DATASET

**Superior Performance of DOCREWARD over Baselines.** As presented in Table 2, on the human-annotated evaluation dataset, DOCREWARD-3B and DOCREWARD-7B, achieve substantial improvements over strong baselines including GPT-4o, Claude Sonnet 4, and GPT-5. In particular, DOCREWARD-7B achieves an overall human preference accuracy of 89.22% , 19.45 points higher than the strongest closed-source baseline (GPT-5, 69.77%). In the critical "Real vs. Synth" setting, DOCREWARD-7B achieves 97.42%, indicating near-perfect alignment with human judgments when distinguishing professional human-authored documents from synthetic ones. Even in the more challenging "Synth vs. Synth" setting, DOCREWARD-7B maintains 78.22%, substantially higher than GPT-5 (64.85%). These results demonstrate that DOCREWARD effectively captures structural and stylistic quality signals that existing LLMs overlook.

### 4.4 IMPROVING DOCUMENT GENERATION WITH DOCREWARD

To demonstrate how DOCREWARD improves document generation, we conduct two complementary experiments.

**DOCREWARD is used as a re-ranking model for multiple rollouts.** A document agent generates $N$ documents given the same text content, and then a reward model identifies the best one from the documents according to their scores. We compare three reward models: random, GPT-5, and DOCREWARD. Human annotators rank the selected documents from each reward model according to their structure and style. Finally, we calculate the win/lose/tie rates for each reward model against the others. As presented in Table 3, the random baseline performs poorly, winning only 24.6% of comparisons and losing 66.2%. GPT-5 achieves more balanced results with a win rate of 37.7%. By contrast, DOCREWARD substantially outperforms both baselines, achieving a win rate of 60.8%

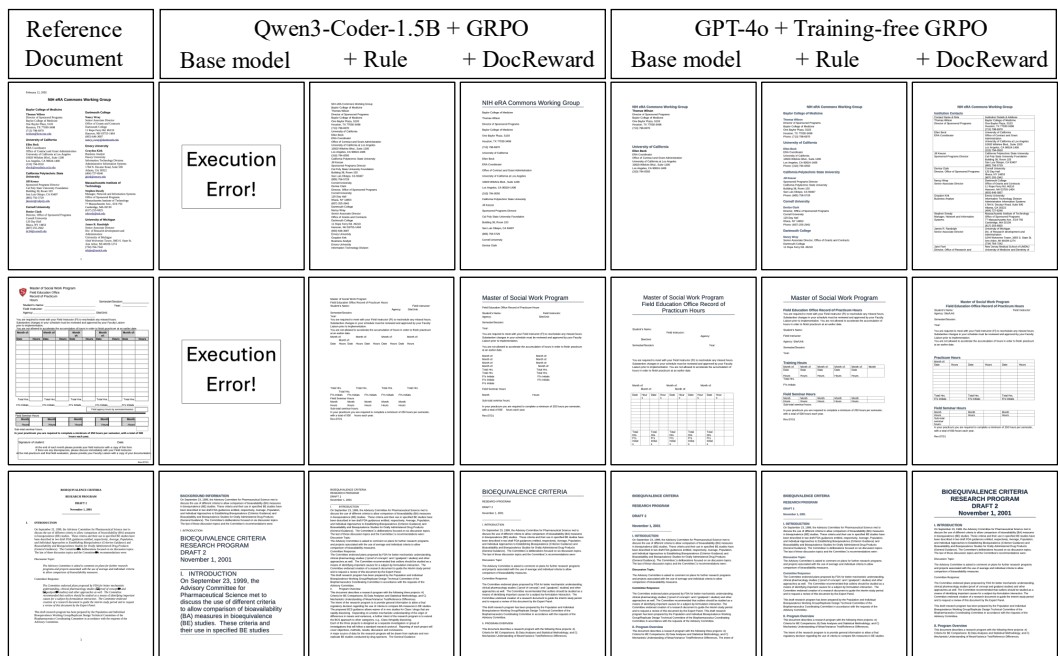

Figure 5: Visualization of generated documents reinforced by DOCREWARD.

and losing only $16.9\%$ of the time. These results indicate that DOCREWARD's reward signal better captures the structural and stylistic qualities that humans value. The evaluation demonstrates that plugging DOCREWARD into a standard document agent improves the output document without changing the underlying agent. The evaluation details are presented in Appendix A.7.

**RL via DOCREWARD enhances open- and closed source agentic workflows.** We aim to enhance document generation agentic workflows that take plain text as input and generate professional Word document. We consider two kinds of rewards: 1) $\mathcal{R}_{\texttt{rule}}$ that penalizes documents that either result from invalid `python-docx` code or differ from the input text after execution. Specifically, if the code executes successfully, then $\mathcal{R}_{\texttt{rule}} = ROUGE(doc\_ori, doc\_gen)$; else $\mathcal{R}_{\texttt{rule}}$ will be zero. 2) $\mathcal{R}_{\texttt{DocReward}}$ that penalizes documents with poor structure and style. Overall, the reward assigned to each generated document is defined as:

$$\mathcal{R}_{\texttt{rule}} + \alpha \cdot \mathbb{I}_{\texttt{rule}} \cdot \sigma(\mathcal{R}_{\texttt{DocReward}}), \tag{3}$$

where $\alpha$ is a hyperparameter to balance the rewards, $\sigma(\cdot)$ is the Sigmoid operation to regularize the value range of DOCREWARD to $(0, 1)$, and $\mathbb{I}_{\texttt{rule}}$ represents whether $\mathcal{R}_{\texttt{rule}}$ is larger than a threshold.

For open-source models, we adopt GRPO (Shao et al., 2024) as the reinforcement learning algorithm, while employing training-free GRPO (Cai et al., 2025) for closed-source models. After RL, human annotators are asked to rank the documents produced by six model variants. The evaluation criterion is the professionalism of the document's structure and styling. The experimental results are shown in Table 4. Rule-based rewards substantially improve the success rate of document generation for both Qwen2.5-Coder and GPT-4o. Incorporating DocReward as reward further enhances the performance of both open- and closed-source models, leading to higher success rates, improved ROUGE-L scores, and better average rankings. These results demonstrate that DOCREWARD serves as an effective reward model for professional structure and style. Figure 5 presents visualization of documents generated by different models.

## 4.5 CASE STUDY

We present a case study on documents with identical textual content but different structures and styles in Figure 6. In case (a), the allocation of whitespace is ineffective, with insufficient space

| | Success Rate↑ | ROUGE-L↑ | DocReward↑ | Rank↓ |
|---|---|---|---|---|
| Qwen2.5-Coder | 30.0 | 20.61 | 0.0663 | 4.58 |
| - w/ GRPO (rule) | 98.0 | 97.94 | 0.1785 | 4.06 |
| - w/ GRPO (rule&DocReward) | 100.0 | 97.95 | 0.3046 | 2.84 |
| GPT-4o | 52.0 | 48.73 | 0.2682 | 3.18 |
| - w/ Training-free GRPO (rule) | 66.0 | 62.15 | 0.3189 | 2.70 |
| - w/ Training-free GRPO (rule&DocReward) | 78.0 | 74.33 | 0.4486 | 2.02 |

Table 4: Results of the reinforcement learning experiments. "DocReward" denotes the sigmoid-normalized DocReward score. "Rank" denotes the average ranking assigned by human annotators.

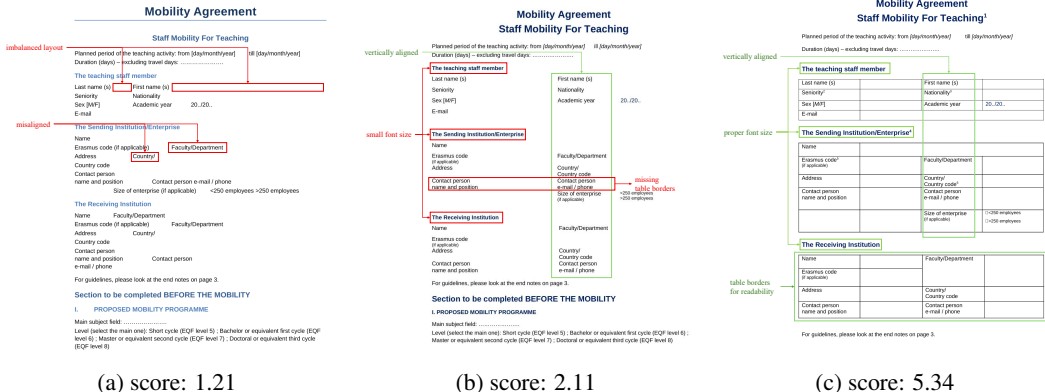

(a) score: 1.21    (b) score: 2.11    (c) score: 5.34

Figure 6: Case study: DOCREWARD's scores reflect structural and stylistic professionalism.

between *Last Name* and excessive space between *First Name*, leading to an imbalanced layout. Key fields such as *Faculty/Department*, *Country*, and *Country Code* are not vertically aligned, causing a cluttered and disorganized layout. This poor alignment and inconsistent spacing result in a low score of 1.21 from DOCREWARD. Case (b) adopts a table-like arrangement, but the level-1 heading *The teaching staff member* is too small and does not stand out from the body text, diminishing its impact. Additionally, the lack of borders around input fields makes it hard to locate items easily, resulting in a moderate score of 2.11. Case (c) provides a clear and well-structured layout, with headings appropriately larger than the body text and better readability, earning the highest rating of 5.34. These results show that DOCREWARD effectively captures document professionalism in structure and style. Additional cases are provided in Appendix A.9.

## 4.6 VISUALIZATION OF ATTENTION MAP

To understand DOCREWARD's internal decision-making process, we conduct probing experiments analyzing its attention maps within the language model part. The attention maps are computed over image patches. As shown in Figure 7, the attention maps reveal that the model relies more on structural and formatting cues than on semantic content when evaluating document professionalism. In Figure 7a, attention is focused on headings and numbering, indicating sensitivity to structure clarity and logical flow. The model also allocates considerable attention to page headers (i.e., "CS-66") and footers at bottom right corner (i.e., "DEC. 2006"), suggesting that the inclusion of page headers and footers is an important signal of professional structure. In Figure 7b, the model attends strongly to bullet points, suggesting that formatting consistency and emphasis markers are key professionalism signals. In Figure 7c, attention is dispersed across table grids, highlighting the importance of text alignment and readability in structured tabular layouts. Moreover, the attention maps show notable focus on the four page corners, suggesting that DOCREWARD implicitly checks for uniform margins and balanced whitespace, which are strong indicators of professional layout design.

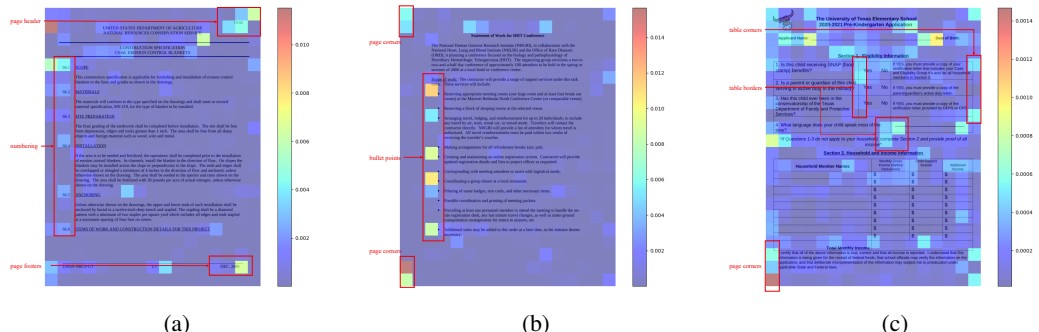

Figure 7: Visualization of attention maps. DOCREWARD captures structural and stylistic elements, such as headings, alignment, and whitespace, in its evaluation of document professionalism.

## 5 RELATED WORK

**Aesthetic and Professionalism Assessment.** In graphic design, AesthetiQ (Zhang et al., 2024) utilizes multimodal LLMs as preference evaluators to align layout generation with aesthetic requirements, while diffusion-based methods such as LACE (Li et al., 2023) introduce differentiable constraints to directly optimize layout attributes. For web and mobile interfaces, systems like Calista (Yu et al., 2019) and Android UIs (Fu et al., 2024) use explicit ratings and pairwise comparisons to model visual appeal, showing correlations with usability. Additionally, photo aesthetics are modeled using layout-aware CNNs such as A-Lamp (Li et al., 2018), and similar techniques extend to video (Liu & Yu, 2023). These studies show that aesthetic principles can guide AI development and that human preferences are reliable supervisory signals, but they focus on images or UI interfaces rather than multi-page documents, where professionalism depends on both structure and style.

**Document AI.** Document AI research mainly targets semantic parsing and content understanding. Models such as LayoutLM (Xu et al., 2020) and ReLayout (Jiang et al., 2024), along with OCR-based pipelines (Subramani et al., 2020), identify logical elements such as headings, tables, and semantic groups to support information extraction and classification. Recent work also explores automatic document or layout generation (Lin et al., 2023; Tang et al., 2023; Tian et al., 2025), but evaluation has primarily been limited to content correctness or basic formatting. As a result, the assessment of document professionalism—particularly visual structure and style—remains largely unexplored.

**Preference Learning and Reward Models.** A major challenge in professionalism assessment is acquiring feedback signals that reflect human judgment. Preference-based reward modeling addresses this issue by training on pairwise comparisons to approximate preferences, forming the basis of alignment methods like RLHF (Stiennon et al., 2020) and DPO (Rafailov et al., 2023). This demonstrates that preference data offers a scalable and effective way to align generative models with nuanced expectations.

## 6 CONCLUSION

In this paper, we introduced DOCREWARD, a Document Reward Model designed to assess structural and stylistic professionalism. Our key contributions include the construction of a multi-domain dataset DOCPAIR of 117K paired documents, each with high- and low-professionalism counterparts. We train DOCREWARD using the Bradley-Terry loss. Rigorous evaluation on a human-annotated test set demonstrated DOCREWARD's superior performance, outperforming GPT-4o and GPT-5 by **30.6**, **19.4** percentage points, respectively in human preference accuracy. Moreover, a human preference evaluation demonstrates its utility to guide generation agents toward producing human-preferred documents.

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

# Table of Contents in Appendix

# A APPENDIX

## A.1 THE USE OF LARGE LANGUAGE MODELS

Following the completion of the draft by the human authors, a large language model was employed to enhance the clarity and academic tone of specific sections.

## A.2 MODEL IMPLEMENTATION DETAILS

Our document reward model is built upon the Qwen2.5-VL multimodal architecture, with the maximum input pixels set to 300,000. It is configured with a maximum context length of 16,000 tokens to ensure comprehensive understanding. Training utilizes the AdamW optimizer with a learning rate of 1e-6 and a batch size of 256 over 3 epochs. All training was conducted on 8 NVIDIA A100 GPUs. The training code is based on LLaMA-Factory (Zheng et al., 2024).

## A.3 SOURCE DOCUMENTS EXPANSION

To ensure that the reward model learns to assess differences in structure and style rather than content, we applied a rigorous filtering process. Using `python-docx`, we extracted text from pairs of Microsoft Word DOCX documents and computed their word counts. Only synthetic documents with a word count difference of no more than 20 words from the original document and a ROUGE-L score exceeding a threshold are retained, ensuring comparable content while isolating variation in structure and style. For the constructed training dataset DOCPAIR, both GPT-4o and GPT-5 serve as the base models of agents.

## A.4 ANNOTATION PROTOCOL AND RELIABILITY

**Annotation Guidelines.** The annotation guidelines consist of general principles that are formulated in an explicit, objective manner. For instance, extremely narrow margins that produce an almost fully saturated page layout are commonly regarded as unprofessional across different cultural and regional contexts. The detailed guideline for human annotation are presented in Figure 8.

**Independence from annotators' cultural and professional backgrounds.** The annotation was performed by three Ph.D. students from diverse fields (computer science, marketing, and mathematics). We measured inter-annotator reliability using Cohen's Kappa; the results are shown in Table 5. The high agreement indicates that the annotations follow clear, well-defined rules that do not depend on the annotators' professional training or cultural background, demonstrating the guidelines' generality and objectivity.

## A.5 GENERALIZATION ABILITY OF DOCREWARD

**Out-of-Domain Evaluation.** Table 6 reports the in-domain and out-of-domain results across different models. Firstly, DocReward-7B (85.55) remains superior to all baseline models, including the closed-source model GPT-5 (71.11). This trend is consistent with the in-domain results. Secondly, The performance of DocReward-7B decreases by merely 3.67 percentage points when transitioning from in-domain to out-of-domain evaluations. Such a small performance gap indicates that DocReward generalizes effectively to unseen domains.

Table 5: Pairwise Cohen's Kappa among annotators.

|  | Annotator 1 | Annotator 2 | Annotator 3 | Average |
|---|---|---|---|---|
| **Annotator 1** | - | 83.40 | 80.92 | 82.15 |
| **Annotator 2** | 83.40 | - | 85.90 | 84.65 |
| **Annotator 3** | 80.92 | 85.90 | - | 83.41 |
| **Average** | 82.15 | 84.65 | 83.41 | 83.40 |

```
Guidelines for Human Annotation

Target:
Each document group contains N documents. Their textual content is the
    same, but their structure and style differ. The first document in
    each group is the original human-authored document, which serves
    as a reference during annotation. Based on the level of
    professionalism in structure and style across the N documents, the
    annotator should rank the documents. Note that there may exist
    cases where human-authored documents are not the best one.

Annotation Format Example:
For example, for the document group with ID 10655307, suppose the
    human-annotated professionalism ranking is 1 > 5 > 3 > 2 > 4,
    where 1 is judged to be the most professionally structured and
    formatted document, and 4 is the least professional. Then the
    annotation format should be: 10655307 \t 15324

Evaluation Criteria:

1. **Layout and Design**:
   - Consistent formatting and spacing

   - Proper use of headings, subheadings, and other structures, and
     proper hierarchy (e.g., long paragraphs should use body text style
     rather than heading styles, and headings should not be formatted
     as body text)
   - Appropriate margins and white space usage (e.g., page margins or
     table column widths that are excessively wide or narrow are not
     appropriate)
2. **Readability and Typography**:
   - Consistent and appropriate font choices
   - Proper Text size (e.g., overly large or overly small text is not
     suitable)
   - Appropriate line spacing and clear paragraph structure
   - Proper Text alignment
3. **Professional Standards**:
   - Document structure and organization
   - Use of professional elements (headers, footers, page numbers)
   - Consistency across pages (if multiple pages provided)
4. **Visual Elements**:
   - Quality and placement of images, tables, or charts
   - Integration of visual elements with text
   - Professional presentation of data
```

Figure 8: Detailed guideline for human annotation.

Table 6: In-domain and out-of-domain performance.

| Model | In-domain | Out-of-domain |
|---|---|---|
| Qwen2.5 VL-3B | 34.88 | 31.10 |
| Qwen2.5 VL-7B | 50.95 | 45.92 |
| GPT-4o | 58.56 | 57.04 |
| GPT-5 | 69.77 | 71.11 |
| DocReward-3B | 86.89 | 81.85 |
| DocReward-7B | 89.22 | 85.55 |

Table 7: Cross-lingual robustness evaluation across multiple languages. "Non-English" denotes the average performance on documents written in non-English languages.

| | French | Spanish | Rusian | Non-English Avg. | English | Drop |
|---|---|---|---|---|---|---|
| Qwen2.5 VL-3B | 35.00 | 33.75 | 22.50 | 30.42 | 34.88 | -4.46% |
| Qwen2.5 VL-7B | 48.75 | 42.50 | 27.50 | 39.58 | 50.95 | -11.37% |
| GPT-4o | 47.50 | 52.50 | 42.50 | 47.50 | 58.56 | -11.06% |
| GPT-5 | 57.50 | 76.25 | 47.50 | 60.42 | 69.77 | -9.35% |
| DocReward-3B | 77.50 | 82.50 | 72.50 | 77.50 | 86.89 | -9.39% |
| DocReward-7B | 78.75 | 88.75 | 66.25 | 77.90 | 89.22 | -11.32% |

**Cross-Lingual Robustness:** To evaluate robustness across languages, we conduct experiments in French, Spanish, and Russian. The results are as shown in Table 7. Firstly, the DocReward-7B model achieved a high score of 77.90, substantially outperforming all baseline models (exceeding GPT-5 (60.42) by 17.48 percentage points). This is consistent with the conclusions drawn from the English evaluation. Secondly, all models, including the baselines, exhibited performance degradation in non-English settings. For example, GPT-5 dropped by 9.35%. The performance drops of DocReward ($-9.39\%$, $-11.32\%$) are comparable to those of the closed-source models GPT-4o ($-11.06\%$) and GPT-5 ($-9.35\%$), indicating that DocReward demonstrates strong cross-lingual robustness.

## A.6 PROMPTS

---

**Domain and Type Classification Prompt**

```
You are an expert document quality evaluator and domain classifier.
    Your task is to assess the professionalism, layout quality, and
    readability of documents based on their visual appearance, and
    classify the document's domain.

You will be provided with screenshot images of document pages. First,
    classify the document domain and then evaluate the document on
    quality criteria.

**DOMAIN AND DOCUMENT TYPE CLASSIFICATION**:
Classify the document on two levels:

1. **Domain Classification**: Choose the broad domain category (e.g.,
    technical, personal, legal, scientific, government, financial,
    medical, business, education, marketing, academic, news,
    entertainment, sports, non_profit, religious, insurance,
    real_estate, automotive, travel, hospitality, retail,
    manufacturing, logistics, etc.)

2. **Document Type Classification**: Identify the specific document
    type within that domain. Examples include:
    - Technical: engineering_report, user_manual,
    software_documentation, specification_document, etc.
    - Personal: cv, personal_report, resume, personal_letter, etc.
    - Legal: legal_brief, legal_opinion, contract, regulatory_text,
    court_filing, etc.
    - Scientific: technical_paper, research_publication,
    scientific_study, laboratory_report, etc.
    - Government: regulation, white_paper, official_report,
    government_form, policy_document, etc.
    - Financial: audit_report, investment_report, financial_statement,
    banking_document, etc.
    - Medical: pharmaceutical_document, clinical_report,
    medical_manual, research_study, etc.
```

```
   - Business: corporate_memo, business_plan, presentation,
    financial_report, marketing_brochure, etc.
   - Education: thesis, textbook, academic_report, research_paper,
    course_material, etc.
   - Marketing: brand_guidelines, campaign_brief,
    advertising_proposal, market_analysis, social_media_strategy, etc.
   - Academic: dissertation, grant_proposal, conference_paper,
    journal_article, literature_review, etc.
   - News: press_release, news_article, interview_transcript,
    editorial, media_kit, etc.
   - Entertainment: production_notes, script, event_program,
    casting_call, performance_review, etc.
   - Sports: athlete_profile, game_report, coaching_guide,
    training_manual, tournament_bracket, etc.
   - Non_profit: annual_report, fundraising_proposal, impact_report,
    volunteer_handbook, grant_application, etc.
   - Religious: ceremony_program, sermon_notes, prayer_book,
    religious_text, pastoral_letter, etc.
   - Insurance: claims_form, policy_document, underwriting_report,
    risk_assessment, coverage_summary, etc.
   - Real_estate: lease_agreement, property_listing, market_analysis,
    appraisal_report, property_brochure, etc.
   - Automotive: parts_catalog, service_manual, recall_notice,
    safety_report, warranty_document, etc.
   - Travel: travel_guide, itinerary, visa_application,
    booking_confirmation, hotel_brochure, etc.
   - Hospitality: staff_handbook, menu, guest_services_guide,
    reservation_system, event_planning_document, etc.
   - Retail: inventory_report, product_catalog, customer_survey,
    sales_analysis, store_policy, etc.
   - Manufacturing: production_schedule, quality_control_report,
    equipment_manual, safety_protocol, process_documentation, etc.
   - Logistics: delivery_schedule, shipping_manifest,
    transportation_plan, warehouse_inventory, supply_chain_analysis,
    etc.

Choose the most specific and accurate document type that describes the
    document's purpose and content. You may use other document types
    not listed above if they better describe the document.
```

Document Scoring Prompt for Proprietary Models (point-wise)

```
You are an expert document quality evaluator. Your task is to assess
    the professionalism, layout quality, and readability of documents
    based on their visual appearance.

You will be provided with screenshot images of document pages.
    Evaluate the document on the following criteria:

1. **Layout and Design**:
   - Professional appearance and visual appeal
   - Consistent formatting and spacing
   - Proper use of headings, subheadings, and hierarchy
   - Appropriate margins and white space usage
   - Overall visual balance and organization

2. **Readability and Typography**:
   - Font choices and consistency
   - Text size and legibility
   - Line spacing and paragraph structure
   - Text alignment and justification
```

```
3. **Professional Standards**:
   - Document structure and organization
   - Use of professional elements (headers, footers, page numbers)
   - Consistency across pages (if multiple pages provided)
   - Overall polish and attention to detail

4. **Visual Elements**:
   - Quality and placement of images, tables, or charts
   - Integration of visual elements with text
   - Professional presentation of data

Rate the document on a scale from 0 to 10, where:
- 9 to 10: Exceptional professional quality
- 7 to 8: High professional standard
- 5 to 6: Good professional appearance
- 4: Average / acceptable quality
- 2 to 3: Below average, needs improvement
- 0 to 1: Poor quality, significant issues

Your response should follow this format:
1. First, provide a detailed analysis of each evaluation criteria
   mentioned above
2. Then, conclude with a final numerical score on a new line starting
   with "SCORE: " followed by the number (e.g., "SCORE: 7.250")
```

### Document Scoring Prompt for Proprietary Models(Pair-wise)

```
You are an expert document quality evaluator. Your task is to compare
    two documents and determine which one has better professionalism,
    layout quality, and readability based on their visual appearance.

You will be provided with screenshot images of all pages from two
    documents: Document A and Document B. Compare the documents on the
    following criteria:

1. **Layout and Design**:
   - Professional appearance and visual appeal
   - Consistent formatting and spacing
   - Proper use of headings, subheadings, and hierarchy
   - Appropriate margins and white space usage
   - Overall visual balance and organization

2. **Readability and Typography**:
   - Font choices and consistency
   - Text size and legibility
   - Line spacing and paragraph structure
   - Text alignment and justification

3. **Professional Standards**:
   - Document structure and organization
   - Use of professional elements (headers, footers, page numbers)
   - Consistency across pages
   - Overall polish and attention to detail

4. **Visual Elements**:
   - Quality and placement of images, tables, or charts
   - Integration of visual elements with text
   - Professional presentation of data

Your response should follow this format:
1. First, provide a detailed comparative analysis of each evaluation
   criteria for both documents
```

```
2. Then, conclude with your preference on a new line starting with
   "PREFERENCE: " followed by either "A" or "B" (e.g., "PREFERENCE:
   A", "PREFERENCE: B")

Choose the document that demonstrates superior overall quality,
   professionalism, and visual presentation.
```

**Document Scoring Prompt for Proprietary Models (triple-wise)**

```
You are an expert document quality evaluator. Your task is to compare
   two documents and determine which one has better professionalism,
   layout quality, and readability based on their visual appearance.

You will be provided with screenshot images of all pages from three
   documents: Document A, Document B, and the Original document
   (ground truth reference). The Original document serves as a
   reference standard. Compare Documents A and B on the following
   criteria:

1. **Layout and Design**:
   - Professional appearance and visual appeal
   - Consistent formatting and spacing
   - Proper use of headings, subheadings, and hierarchy
   - Appropriate margins and white space usage
   - Overall visual balance and organization

2. **Readability and Typography**:
   - Font choices and consistency
   - Text size and legibility
   - Line spacing and paragraph structure
   - Text alignment and justification

3. **Professional Standards**:
   - Document structure and organization
   - Use of professional elements (headers, footers, page numbers)
   - Consistency across pages
   - Overall polish and attention to detail

4. **Visual Elements**:
   - Quality and placement of images, tables, or charts
   - Integration of visual elements with text
   - Professional presentation of data

Your response should follow this format:
1. First, provide a detailed comparative analysis of each evaluation
   criteria for both documents, taking the Original document as
   reference for quality standards
2. Then, conclude with your preference on a new line starting with
   "PREFERENCE: " followed by either "A" or "B" (e.g., "PREFERENCE:
   A", "PREFERENCE: B")

Choose the document that demonstrates superior overall quality,
   professionalism, and visual presentation.
```

**Prompt for Document Generation**

```
Based on the following plain text content (extracted from a DOCX
   document), generate Python code using python-docx library to
   create a new, well-formatted DOCX document with appropriate styles
   and formatting:

Plain Text Content (no formatting):
```

```
{editing_plan}

Output file: {output_file_path}

TASK OVERVIEW:
You are given ONLY the plain text content of a document (without any
    formatting, styles, or structure information). Your job is to:
1. Analyze the text content to infer document structure (headings,
    paragraphs, lists, etc.)
2. Create a new DOCX document from scratch
3. Apply appropriate professional formatting and styles to make it
    look like a proper document
4. Add visual hierarchy, consistent formatting, and professional
    appearance

IMPORTANT REQUIREMENTS:
1. Create a completely NEW DOCX document based on the plain text
    content
2. **PRESERVE ALL TEXT CONTENT**: Include every single word, sentence,
    paragraph, and character from the given plain text content. Do NOT
    omit, skip, or modify any text content.
3. **NO CONTENT CHANGES**: Only infer and apply formatting/structure.
    The actual text content must remain exactly the same as provided.
4. Analyze the text content to infer document structure and apply
    appropriate formatting
5. Generate Python code that creates a professional-looking document
    with proper hierarchy and styling
6. Ensure ALL provided text appears in the final document in the
    original order
7. **YOUR CODE WILL BE EXECUTED**: The generated Python code will be
    run directly, so it must be complete, executable, and include the
    document.save() function to save the DOCX file to the specified
    output path.
8. **DO NOT USE PLACEHOLDERS OR OMITTED CODE**: The generated code
    MUST be complete and explicit. Do NOT use comments or placeholders
    such as "# ... (Continue to add other sections and paragraphs
    similarly)" or "# Add more content here". The code must include
    ALL content from the original plain text, fully processed and
    added to the document.

**OUTPUT PATH REQUIREMENTS:**
- You MUST use the exact output path provided: {output_file_path}
- DO NOT create your own filename or path
- DO NOT save to current directory with arbitrary names like
    'output.docx', 'document.docx', etc.
- DO NOT use variables like 'output_path' without setting them to the
    exact provided path

CODE STRUCTURE REQUIREMENTS:
Your generated Python code must follow this EXACT structure:

```python
import os
from docx import Document
from docx.shared import Inches, Pt
from docx.enum.text import WD_ALIGN_PARAGRAPH
from docx.enum.style import WD_STYLE_TYPE
# Add other imports here...

# Create new document
doc = Document()

# Add content here with appropriate formatting
```

```
# Process the text content and add to document...

# Create output directory if needed
os.makedirs(os.path.dirname(output_file_path), exist_ok=True)
try:
    print('CODE: output_file_path = ', output_file_path)
except:
    print('CODE: output_file_path ERROR! ')
doc.save(output_file_path)
```

## Prompt for Document Refinement (Phase 1 - Plan Generation)

You are a document formatting analysis expert. Your task is to analyze
    the differences between a previously generated document and the
    ground truth document, then create a specific refinement plan.

**Input Information:**

**1. Previous Generated Code:**
```python
{previous_code}
```

**2. Previous Generated Document Screenshot:**
{previous_doc_screenshot_info}

**3. Ground Truth Document Screenshot:**
{gt_screenshot_info}

**4. Ground Truth Document Representation:**
```
{gt_doc_repr}
```

**Important Context Limitations:**
Due to input context length constraints, the Ground Truth Document
    Representation, Ground Truth Document Screenshot, and Previous
    Generated Document Screenshot may only contain the initial/front
    portions of the documents. However, the Previous Generated Code is
    complete and contains the full implementation. When analyzing
    differences, focus primarily on the visible portions but consider
    that the documents may extend beyond what is shown.

**Task:**
Compare the previous generated document with the ground truth
    document. Identify the 5 most important differences and create a
    specific, actionable refinement plan with concrete implementation
    details needed to modify the previous generated code.

**Output Format:**
Provide a detailed refinement plan with specific values and
    implementation details:

## Top 5 Key Differences and Improvements Needed:

For each improvement, specify:
1. **Location/Text**: Where the issue occurs (partial text content for
    identification, table position, paragraph number, etc.)
2. **What needs to be changed** (exact element/section)
3. **Current state** (what the code currently does)
4. **Target state** (what it should be)

```
5. **Specific implementation** (exact font sizes, spacing values,
    alignment settings, etc.)

### Example format:
**Issue**: [Specific formatting problem]
- **Location**: Text containing "Document Header" or Table in section
    2, row 1
- **Current**: Font size 12pt, left alignment
- **Target**: Font size 14pt, center alignment
- **Implementation**: Set `run.font.size = Pt(14)` and
    `paragraph.alignment = WD_ALIGN_PARAGRAPH.CENTER`

**Issue**: [Table formatting problem]
- **Location**: Table with headers "Product Name, Price"
- **Current**: No borders, default spacing
- **Target**: 1pt black borders, 6pt cell padding
- **Implementation**: Add table border properties with `width=1pt,
    color=black` and set cell margins to `6pt`

Focus on providing exact values (font sizes in pt, spacing in
    pt/inches, specific color values, alignment constants) and
    concrete python-docx implementation steps. **Limit to exactly 5
    most important differences** that will have the biggest visual
    impact.
```

Prompt for Document Refinement (Phase 2 - Code Generation)

```
You are a document generation expert. Your task is to generate
    improved Python code that addresses the specific formatting issues
    identified in the refinement plan.

**Input Information:**

**1. Previous Generated Code:**
```python
{previous_code}
```

**2. Refinement Plan:**
```
{refinement_plan}
```

**3. Output File Path:**
- Output file: {output_file_path}

**Task:**
Based on the previous code and the refinement plan, generate a
    **complete and improved Python code** that creates a document
    matching the ground truth as closely as possible. This should be a
    standalone, executable script that generates the entire document
    from scratch.

**Requirements:**
1. **Generate complete Python code** – not just modifications, but a
    full working script
2. **Apply all improvements** specified in the refinement plan
3. **Create the entire document** structure and content to match
    ground truth
4. **Use appropriate libraries** (python-docx for high-level
    operations, direct XML manipulation for precise control)
5. **Include error handling** for robustness
```

```
6. **Save to specified output path** – the code must generate a
   complete document file
7. **DO NOT use main() function wrapper** – code should execute
   directly at top level
8. **Use exact output path provided**: {output_file_path}

**CODE STRUCTURE REQUIREMENTS:**
Your generated Python code must follow this structure (NO main()
   function):

```python
import os
from docx import Document
from docx.shared import Inches, Pt
from docx.enum.text import WD_ALIGN_PARAGRAPH
# Add other imports as needed...

# Create new document
doc = Document()

# Add all content here with appropriate formatting
# Apply all improvements from refinement plan...

# Save the document
output_file_path = "{output_file_path}"
os.makedirs(os.path.dirname(output_file_path), exist_ok=True)
doc.save(output_file_path)
print("CODE: output_file_path = ", output_file_path)
```

**Advanced Formatting Capabilities:**
- **python-docx API**: Use for standard document operations
- **Direct XML manipulation**: Use when python-docx doesn't provide
   sufficient control
  – Access underlying XML: `element._element`
  – XPath queries: `element.xpath()`
  – Direct attribute setting: `element.set()` on XML nodes
  – Namespace operations: Use `qn()` for proper namespace handling
  – Document XML access: `document.element.body` for document-level
   changes

**Code Structure:**
The code should be a complete script that:
- Creates a new document
- Builds the entire document structure and content
- Applies all formatting to match the ground truth
- Saves the complete document to output_file_path

**Output Format:**
Provide a complete, executable Python script that implements the
   improvements specified in the refinement plan.

**XML Manipulation Reference:**
When python-docx API is insufficient, you can use direct XML
   manipulation. Here are helper functions and examples for reference:

*Helper functions (include only if needed):*
```python
def set_xml_attribute(element, attr_name, attr_value):
    """Set XML attribute directly on element"""
    if hasattr(element, '_element'):
        element._element.set(qn(attr_name), attr_value)
    else:
```
```

```
        element.set(qn(attr_name), attr_value)

def add_xml_element(parent, tag_name, **attributes):
    """Add XML element with attributes"""
    element = OxmlElement(qn(tag_name))
    for attr, value in attributes.items():
        element.set(qn(attr), value)
    parent.append(element)
    return element
```

*Example XML operations:*
- For precise spacing control: `p_element = paragraph._element;
    spacing_element = add_xml_element(p_element, 'w:spacing',
    before="120", after="120")`
- For table borders: `table_element = table._element; table_props =
    add_xml_element(table_element, 'w:tblPr')`
- For direct attribute setting: `element._element.set(qn('w:val'),
    'value')`

**Focus on:**
- Precise implementation of the refinement plan using both python-docx
    API and direct XML manipulation
- Proper python-docx syntax and XML node manipulation for fine-grained
    control
- Maintaining document integrity while applying improvements
- Clear, maintainable code structure with comprehensive error handling
- Complete document generation (not just partial modifications)

## A.7 DETAILS OF EXTRINSIC EVALUATION

The *Textual Content to Document* defined in Section 3.1 is adopted as the document agent, with the base model being GPT-5. Three reward models, including random, GPT-5, and DOCREWARD are compared. Once the document agent generates candidates and the reward model selects the top-ranking document from $N$ candidates, a highly educated annotator is asked to rank the three documents selected, according to the definitions of professional structure and style defined in Figure 8. As a result, documents from each reward model are annotated 130 comparison pairs against those of another reward model. Finally, the win/lose/tie rate of each reward model is calculated on the comparison pairs against the other reward models.

## A.8 ABLATION STUDY OF INPUTS

In designing the input channels for DOCREWARD, we experimented with two different configurations: a purely visual channel method and a combination method of visual and additional parsing information. The experimental results are summarized in Table 8.

| Model | Human Preference Accuracy (%) | | |
| --- | --- | --- | --- |
| | Synth vs. Synth | Real vs. Synth | Overall |
| image-only (3B) | 70.92 | 94.98 | 85.00 |
| image + OCR text & bbox (3B) | $63.13_{(-7.79)}$ | $92.46_{(-2.52)}$ | $80.30_{(-4.7)}$ |
| image-only (7B) | 73.75 | 97.99 | 87.94 |
| image + OCR text & bbox (7B) | $68.08_{(-5.67)}$ | $95.98_{(-2.01)}$ | $84.41_{(-3.53)}$ |

Table 8: Additional text and bounding box of text span are not helpful for the assessment of professional structure and style.

## A.9 MORE EXAMPLES OF CASE STUDY

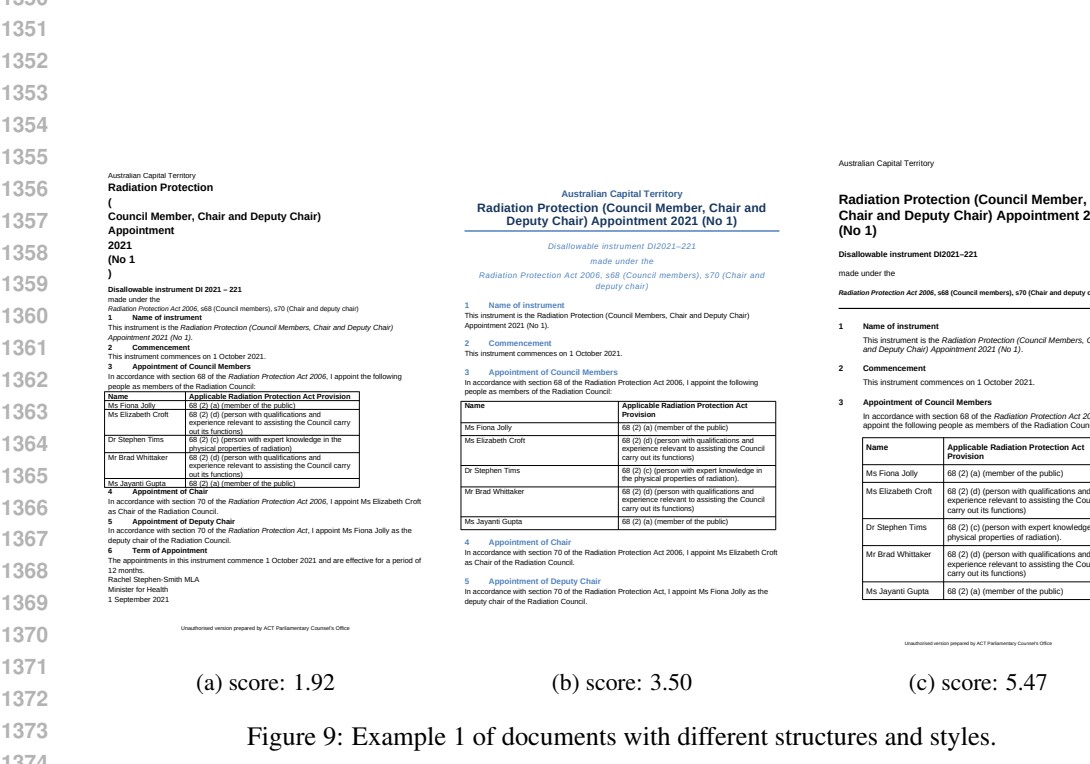

(a) score: 1.92          (b) score: 3.50          (c) score: 5.47

Figure 9: Example 1 of documents with different structures and styles.

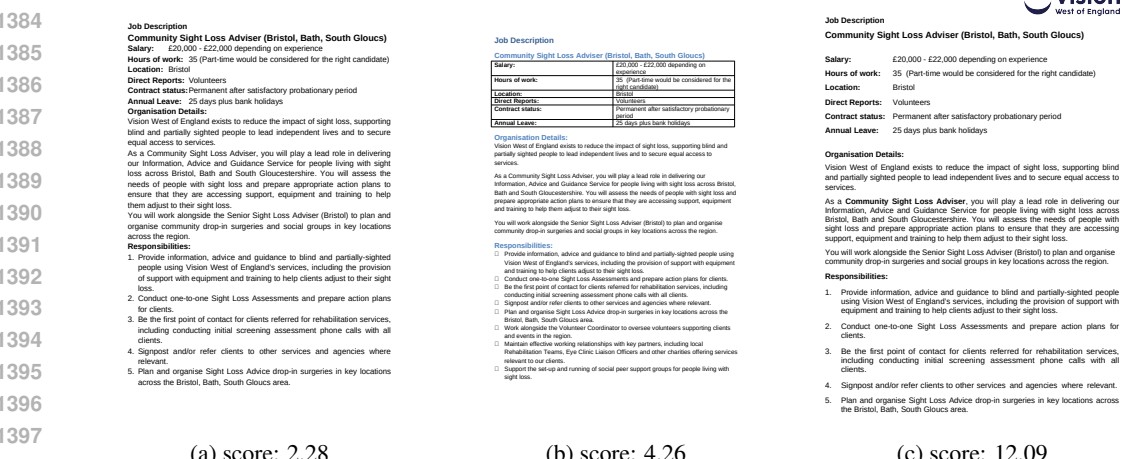

(a) score: 2.28          (b) score: 4.26          (c) score: 12.09

Figure 10: Example 2 of documents with different structures and styles.

