# OpenReview forum: "DocReward: A Document Reward Model for Structuring and Stylizing"
_ICLR.cc/2026/Conference — ICLR 2026 Conference Withdrawn Submission_

### Official Review · Reviewer_XQhC · 2025-10-19

**Soundness:** 2
**Presentation:** 3
**Contribution:** 2
**Rating:** 2
**Confidence:** 4

**Summary:**

The paper describes work on building a reward model for optimizing professional aesthetics of documents for document-generation/document-assessment agentic workflows. The authors define the notion of “professionalism” based on two factors 1) structure pertaining to proper use of spacing, indention, alignment, breaks, etc and 2) style pertaining proper use of fonts, headings, emphasis, numbering, formatting, etc. The authors propose a three-step agent-based data construction pipeline to augment existing professional document datasets (GovDocs, NapierOne, collected docs) and rank based on GPT-5 judgments of professionalism. The final dataset result contains 117K paired documents, covering 32 domains and 267 document types. For the experiments, the author use Qwen-2.5-VL as the model of choice to be optimized and compared its performance exclusively with commercial models (GPT,-4o and 5 and Claude Sonnet) across pairwise and pointwise setups. Results show that DocReward-7B outperforms all other models by some margin but no tests were conducted whether the performance advantage is significant. The authors also miss comparing with strong baselines (vanilla Qwen) as well as strong openweight VL models that could have increased the paper’s empirical findings and technical rigor. There are also concerns on proper capturing of rewards from human preferences conducted by the authors. At this current stage, I’m not confident in accepting the paper at its current state. More information on issues and points for improvement are discussed below.

**Strengths:**

The paper is fairly well written and easy to read. The problem being tackled on notions of professionalism in documents is important but requires proper modelling of desired outcomes or aesthetics from diverse human preferences. The augmented dataset from the paper DocPairs may be of use for the community provided that proper licensing and terms of agreements are clearly-defined.

**Weaknesses:**

The experimented models are restrictive and exclusive to commercial LLMs and lack strong baseline comparisons. Why did you not include vanilla unoptimized Qwen-2.5-VL (3B, 7B, and 32B) for pairwise and pointwise settings? Since you’re using this as your baseline model, it’s perfectly fair to also compare it without the additional task-based optimization for document rewarding. For additional rigor and empirical evaluations to strengthen the paper’s findings, the authors can also add more updated or openweight VLMs such as DeepSeek-VL, CogVLM, etc.

There is very limited information in the human-related preference ranking which makes the paper confusing. How many (skilled) human annotators were asked to rank preferred documents? How are the pairs distributed? If only one person is employed, the reward model might be shortsighted and only favors one person’s notion of professionality. Please clarity/justify this part.

Similar to the previous question, what are the domain backgrounds of the annotators? Human annotators have inherent biases such as an annotator working on education domain might prefer formats from education documents and treat it as the gold-standard professional document which still qualifies for the “professional” definition provided by the authors. How does the study handle this possible bias? What is the specific instruction provided to the annotators when ranking?

I strongly suggest using a more robust and recognized metric like Cohen’s kappa for calculating annotator reliability across all experiments than percentage scoring as this is misleading.

The paper lacks error analysis of the reward models. Instead of the case study, I would prefer learning about cases of instances where say a human rates a certain document layout as high score but the reward model rates it with a low score (and vice versa). These misjudgments are important to diagnoses certain weaknesses of the reward model which can be attributed to factors such as limited diversity of preference ranking, limited variation in document sources (since the domain distribution is not balanced), etc.

There are parts using the phrase “well-educated human evaluators” - Please a more specific wording such as “qualified” or “skilled” (with n number of domain-specific experience if applicable) instead of “well-educated” as this is vague.

**Questions:**

Did you measure for data overlap from the existing public collections of document datasets (GovDocs1 and NapierOne) against the ones you collective via Common Crawl? Both are from public sources so there may be duplicated documents.

What would be the license and terms of agreement that the authors will associate with the new DocPairs dataset? Please mention this explicitly in the paper.

How many (skilled) human annotators were asked to rank preferred documents? How are the pairs distributed?

Are the improvements for models optimized DocReward statistically significant?

---

> ### Author Response · Authors · 2025-11-25
> **Response to reviewer (1/2)**
>
> We thank the reviewer for the detailed and constructive feedback, and we would like to clarify, address the raised concerns, and provide additional context regarding our experiments and dataset.
>
> ---
>
> ### Add More Baselines
>
> 1. The previous paper version did not include comparisons with open-source models such as Qwen-2.5-VL because we found that these open-source models exhibit very **limited** capabilities in evaluating document structure and style. Therefore, we instead used more powerful closed-source models, GPT-5 and Claude Sonnet 4, as **stronger baseline** models for comparison.
> 2. As recommended by the reviewer, we further add experimental results of the mentioned open-source models of comparable model size. As presented in Table 2, the newly added open-source baselines did not surpass the previously reported strongest baseline, GPT-5, whereas DocReward outperformed GPT-5.
>
>
> ---
>
> ### Annotator Number and Background
>
> There were **three annotators** involved in the annotation process, each a Ph.D. student from the Departments of Computer Science, Business, and Mathematics, respectively. Each pair to be annotated was randomly assigned to one of the annotators.
>
> The annotator number is both **reasonable and cost-effective**. We calculated the inter-annotator agreement across annotators from different domain backgrounds and the average Cohen's Kappa is **83.40**. The detailed results are shown in the table below. This high-level inter-annotator agreement indicates that the annotation task is objective and well-defined, and that differences in **domain backgrounds did not lead to substantial variations** in annotations. Therefore, three annotators from distinct domains are sufficient to ensure high-quality annotations.
>
> ---
>
> ### Inter-annotator Agreement and Annotation Guideline
>
> 1. **Domain-Agnostic Definition of Professionalism:** We provide annotators with an explicit and formalized annotation guidelines. These standards focus on **general** and **domain-independent** principles of design, such as consistency of margins, text alignment, appropriate use of whitespace, and uniformity of formatting. For example, **extremely narrow margins** that result in an almost fully saturated page layout are widely regarded as unprofessional, regardless of domain background.
>
> 2. As the following table shows, Cohen's Kappa is **83.40**. As described in [1], this is a quite high agreement. This suggests that the annotations are **reliable**.
>
>    |                 | Annotator 1 | Annotator 2 | Annotator 3 | Average |
>    | --------------- | ----------- | ----------- | ----------- | ------- |
>    | **Annotator 1** | -           | 83.40       | 80.92       | 82.15   |
>    | **Annotator 2** | 83.40       | -           | 85.90       | 84.65   |
>    | **Annotator 3** | 80.92       | 85.90       | -           | 83.41   |
>    | **Average**     | 82.15       | 84.65       | 83.41       | 83.40   |
>
> 3. The annotation guidelines for human annotators are presented in the Appendix A.4 .
>
>
>
> [1] McHugh, Mary L. “Interrater reliability: the kappa statistic.” *Biochemia medica* vol. 22,3 (2012): 276-82.

---

> ### Author Response · Authors · 2025-11-25
> **Response to reviewer (2/2)**
>
> ### About Error Analysis
>
> 1. **Out-of-domain experiment.** According to the reviewer’s suggestion, we conducted additional out-of-domain experiments on unseen domains. The results are presented in the table below, from which we draw the following conclusions:
>
>    1. **In the out-of-domain setting, the DocReward-7B model achieves a high score of 85.55**, remaining **superior** to all other baseline models, including the closed-source model GPT-5 (71.11). This trend is consistent with the in-domain results.
>    2. **Superior Generalization of DocReward.** We observe that the DocReward model exhibits only a minimal performance drop when transitioning from in-domain to out-of-domain evaluations. The performance of DocReward-7B decreases by **merely** **3.67** percentage points. Such a small performance gap indicates that DocReward generalizes effectively to **entirely new and unseen** domains.
>
>    |               | In-domain | Out-of-domain |
>    | ------------- | --------- | ------------- |
>    | Qwen2.5 VL-3B | 34.88     | 31.10         |
>    | Qwen2.5 VL-7B | 50.95     | 45.92         |
>    | GPT-4o        | 58.56     | 57.04         |
>    | GPT-5         | 69.77     | 71.11         |
>    | DocReward-3B  | 86.89     | 81.85         |
>    | DocReward-7B  | 89.22     | 85.55         |
>
> 2. **Cross-lingual experiment.** We conducted cross-lingual experiments to evaluate the performance of the DocReward model on non-English documents. The conclusions are as follow:
>
>    1. **Superior Performance of DocReward on Non-English Languages .** The DocReward-7B model still achieved a high score of 77.90, substantially outperforming all baseline models (exceeding GPT-5 (60.42) by 17.48 points). This is consistent with the conclusions drawn from the English evaluation.
>    2. **Robustness in Cross-lingual Settings.** All models, including the baselines, exhibited performance degradation in non-English settings. For example, GPT-5 dropped by 9.35%. The performance drops of DocReward (−9.39%, −11.32%) are **comparable** to those of the closed-source models GPT-4o (−11.06%) and GPT-5 (−9.35%), indicating that DocReward demonstrates strong cross-lingual robustness.
>
> |               | French | Spanish | Rusian | Non-English Avg. | English | Drop    |
> | ------------- | ------ | ------- | ------ | ---------------- | ------- | ------- |
> | Qwen2.5 VL-3B | 35.00  | 33.75   | 22.50  | 30.42            | 34.88   | -4.46%  |
> | Qwen2.5 VL-7B | 48.75  | 42.50   | 27.50  | 39.58            | 50.95   | -11.37% |
> | GPT-4o        | 47.50  | 52.50   | 42.50  | 47.50            | 58.56   | -11.06% |
> | GPT-5         | 57.50  | 76.25   | 47.50  | 60.42            | 69.77   | -9.35%  |
> | DocReward-3B  | 77.50  | 82.50   | 72.50  | 77.50            | 86.89   | -9.39%  |
> | DocReward-7B  | 78.75  | 88.75   | 66.25  | 77.90            | 89.22   | -11.32% |
>
>
>
>
> ---
>
> ### Question about Data Overlap
>
> We perform a deduplication process between our collected document and the existing public collections (GovDocs1 and NapierOne). Specifically, we employed an overlap-based similarity measure, using ROUGE score as the metric, to identify and remove any possible repeated documents.
>
> ---
>
> ### License with DocPair Dataset
>
> Thanks for your kind suggestion about dataset license! Both the newly constructed DocPair dataset and construction pipeline will be open-sourced under MIT license to facilitate further research on document reward model.
>
> ---
>
> ### Statistical Significance
>
> Yes, the improvements is statistically significant. To rigorously verify the statistical significance, we conducted a McNemar test[1] comparing the strongest baseline model, GPT-5, with DocReward. The resulting *p*-value (**≈ 0.000012 < 0.05**) indicates that the difference in predictions between the two models is statistically significant.
>
> [1] Dietterich, Thomas G. "Approximate statistical tests for comparing supervised classification learning algorithms." *Neural computation* 10.7 (1998): 1895-1923.

---

### Official Review · Reviewer_UQBH · 2025-10-30

**Soundness:** 2
**Presentation:** 3
**Contribution:** 3
**Rating:** 4
**Confidence:** 3

**Summary:**

The paper introduces a large-scale document preference dataset and proposes DocReward to evaluate the professionalism of documents. Experiments show that the trained model outperforms existing baselines.

**Strengths:**

1. The paper introduces a large-scale document preference dataset.
2. Experiments demonstrate the effectiveness of the proposed dataset and model.

**Weaknesses:**

1. The proposed dataset and model are limited to Microsoft Word only.

The reason I give a score of 4 and a confidence of 3 is that I have many questions (refer to the questions). Once these questions are addressed, I’m willing to raise my score.

**Questions:**

1.  In Figure 7, why is (c) scored higher than (b)? (I’m not entirely sure about the definition of professionalism. After checking the rules mentioned in Section 4.5, I still prefer (b).)
2. The ranking and prompt do not consider semantic accuracy. How can you ensure that the generated document does not contain incorrect content yet still achieves a high professionalism score due to its layout?
3. The ranking suggests that human-generated documents are always better. However, shouldn’t model-generated documents sometimes outperform human ones? In lines 195–198, the filtering process shows that some low-quality documents are also human-generated. In this case, the reward model may simply learn human style rather than true quality of style. (This is also reflected in the accuracy gap between Synth vs. Synth and Real & Synth.)
4. What if the versions of Microsoft Word are different?

---

> ### Author Response · Authors · 2025-11-25
> **Response to reviewer (1/2)**
>
> We appreciate the reviewer’s thoughtful feedback and the recognition of our dataset. We would like to address the questions raised to clarify our design choices and experimental results.
>
> ---
>
> ### Limitation of Microsoft Word Format
>
> 1. **Model Input and Format Agnosticism.** **DocReward** does not directly process the internal XML structure of Microsoft Word (`.docx`) files. Instead, it evaluates documents based on **rendered page images**. Once a document is rendered as images, its visual layout characteristics become independent of the original file format, whether the source is a PDF, a LaTeX-generated document, a Google Docs export, or a Word file.
> 2. **Rationale for Data Construction Using Word.** To construct the DocPair dataset, it was necessary to generate document pairs with identical textual content but distinct structures and styles. The `.docx` format of Word provides a **programmatic interface**—via `python-docx`—that enables automated manipulation of document structures and styles. This makes the generation process **scalable and controllable**.
> 3. **Comprehensive Document Diversity.** The DocPair dataset includes 117K paired documents spanning **32 domains** and **267 document categories** (e.g., government forms, meeting minutes, course syllabi, technical manuals, etc.). This extensive coverage ensures that the model learns general principles of professional document design rather than relying on Word software.
>
> ---
>
> ### Question about Figure 7
>
> In previous figure 7 (i.e., Figure 9 in revision), compared with document (c), document (b) has the following two shortcomings:
>
> 1. The **column widths** in document (b) are not appropriate; the first column is too wide with excessive empty space compared to the second column. As shown in the annotation guideline in Figure 8, this corresponds to the guideline on **“appropriate margins and white space usage.”**
> 2. The document (b) does not have a **footer**, which is not professional. This corresponds to the guideline on **“use of professional elements (header, footers, page number, etc.)”**

---

> ### Author Response · Authors · 2025-11-25
> **Response to reviewer (2/2)**
>
> ### Question about Semantic Accuracy
>
> 1. **Assurance of Semantic Accuracy.**  When constructing the DocPair dataset, we computed overlap-based similarity (i.e., ROUGE) and differences in word count between the original source documents and the synthesized ones. Prior to ranking, **we removed any synthesized documents whose textual content significantly deviated from that of the original source document**, thereby ensuring the semantic accuracy of the synthesized documents. Moreover, we found that most of removed cases were simply missing later sections of the source document, rather than introducing incorrect content. This shows that the agent performs very well in generating semantically faithful text already.
> 2. **Scope Clarification.** Current LLMs (e.g., GPT-5, Claude, etc. ) are already highly effective at assessing and ensuring both textual quality and semantic accuracy. Our model is designed to address a gap—namely, the limitations of existing LLMs in evaluating visual structure and stylistic aspects. Accordingly, our focus is on optimizing **structure and formatting style** instead of content quality.
>
> ---
>
> ### Question about "human-generated documents are always better"
>
> 1. **Allowing some noise for train-set construction while rigorous human annotation for test-set.** Indeed, there may exist some cases that human-generated document is worse. But the assumption is necessary and proper for large-scale train-set construction. Our preliminary human inspection indicates that current SOTA models (e.g., GPT-5, Claude Sonnet 4) produce documents whose structure and stylistic quality  are inferior to our carefully curated and quality-controlled **human-authored documents**. Therefore, adopting this assumption is both cost-effective and necessary for  large-scale training data construction while maximizing the average data quality. The test set, in contrast, is constructed through strict **human annotation procedures** which can handle the "worse human document" case.
> 2. **Explanation of the accuracy gaps** (Synth vs. Synth vs. Real & Synth)
>    1. **The inherent difficulty of the tasks differs.** Not only DocReward but **all baseline models** also exhibit higher accuracy on Real vs. Synth settings.
>    2. **Under the more challenging *synth-vs-synth* setting,** **DocReward also achieves better than the strongest baseline model GPT-5.** If the reward model were merely learning human patterns rather than the true quality of style, its performance on the synth-vs-synth test set should be comparable to or worse than the baselines. However, on this subset, DocReward achieves a performance of 78.22, substantially exceeding GPT-5 (64.85), which strongly demonstrates that DocReward has **generalized** to capture the **true quality of structure and style**.
>
> ---
>
> ### Question about Word Software Version
>
> We tested different versions of LibreOffice and python-docx and observed that documents generated across different versions did not exhibit significant differences in structure or style. This indicates that the impact of version differences can be ignored.

---

> > ### Comment · Reviewer_UQBH · 2025-11-28
> > **Response to authors**
> >
> > After reading the rebuttal, most of my concerns have been addressed. I also think the cross-lingual experiments significantly strengthen the paper.
> >
> > However, I agree with the other reviewer's opinion that the paper is mostly an application built on an existing pipeline. Its novelty in algorithms is limited. The core contribution lies primarily in the open-sourced artifacts.
> >
> > Based on these points, I am raising my score to 6 and recommending a weak accept.

---

> ### Author Response · Authors · 2025-12-01
> **Response to reviewer**
>
> Thank you for your positive follow-up and for **raising your score to 6**! We sincerely appreciate the recognition of our cross-lingual experiments and the value of the open-sourced artifacts.
>
> We acknowledge that our work leverages existing multimodal model architecture, which is a common and necessary practice in the era of LLMs. However, our core contribution lies not in inventing a new model architecture, but in introducing crucial innovations in task definition, methodology, and evaluation paradigm. Specifically, our advances are as follows:
>
> 1. **Novel Task Formulation**. We formally define, for the first time, the evaluation of documents in terms of their visual **structural and stylistic professionalism,** a key yet underexplored problem. Unlike traditional assessments that focus on the professionalism of textual content  (e.g., grammar, coherence, factual accuracy), our work focuses on the structural and stylistic aspects of professionalism, which is critical for readability and engagement.
> 2. **Content-Agnostic Methodology**. We propose a novel data construction methodology that **decouples the quality of textual content from the professionalism of structure and style**. This is realized through the creation of paired documents where the textual content is identical, but the visual structure and style are varied. This decoupled data construction is a methodological innovation that makes it possible, for the first time, to purely evaluate a document's structure, and style, free from the interference of textual semantic quality.
> 3. **Advances in the Aesthetic Evaluation Paradigm**. Existing aesthetic assessment methods (e.g., AesthetiQ [1], LACE [2]) primarily target images, web pages, or UI interfaces, focusing on **single page** only. In contrast, our work is the first to focus on **multi-page documents**, where evaluation criteria encompass cross-page consistency (e.g., uniform headers/footers, margins, fonts) and complex structural organization (e.g., logical layout, formatting).
> 4. **Missing Piece in Professional Document Creation.** Although current LLMs/agents are good at generating high-quality textual content, they critically lack the necessary visual quality reward signal to optimize their output for structural and stylistic professionalism. This absence leads to generated documents that often fall short in terms of readability and engagement. DocReward is the first dedicated Reward Model designed to fill this critical gap.
>
> [1] Xueru Zhang, et.al. Aesthetiq: Enhancing graphic layout design via aesthetic-aware preference alignment of multi-modal large language models. In The Twelfth International Conference on Learning Representations (ICLR), 2024.
>
> [2] Yuxin Li, et.al. Towards aligned layout generation via diffusion model with aesthetic constraints. In The Eleventh International Conference on Learning Representations (ICLR), 2023.

---

### Official Review · Reviewer_Knxt · 2025-10-31

**Soundness:** 2
**Presentation:** 3
**Contribution:** 2
**Rating:** 4
**Confidence:** 4

**Summary:**

The paper addresses automatic assessment of document professionalism, emphasizing structural and stylistic quality beyond textual content. The authors propose DocReward, a pointwise document reward model that operates on rendered page images and is trained with a Bradley–Terry  preference-learning objective over pairwise comparisons. To enforce textual-quality agnosticism, they construct the DOCPAIR dataset, in which each pair shares identical textual content but differs in structure and style. The model is built on Qwen-2.5-VL with multi-image inputs; a regression head outputs a single scalar score, optimized with the BT loss to separate preferred from non-preferred samples.

**Strengths:**

1. Clear problem specification: The textual content is held fixed and only structure/style are evaluated, thereby avoiding contamination from writing quality or factual correctness; the formal objective is consistent with the annotation protocol.
2. Training data scale and diversity: DOCPAIR spans 32 domains and 267 document types, comprising 117K paired samples, and includes both Real-vs-Synth and Synth-vs-Synth comparisons.
3. Alignment with preference learning: Pairwise supervision with the Bradley–Terry (BT) loss aligns with human-preference data and is consistent with the preference-learning paradigm underlying RLHF/DPO.

**Weaknesses:**

1. The work reads primarily as an engineering integration: dataset construction and a reward-modeling pipeline built on existing multimodal backbones dominate the contribution, while methodological innovations and fundamental advances over existing paradigms (preference learning/layout understanding/aesthetic evaluation) remain unclear.
2. Although the training set covers many document types, it cannot exhaust the long-tail of real-world distributions; the paper does not provide consistent evaluation on unseen types or out-of-domain settings. It is recommended to conduct explicit out-of-domain experiments (ensuring certain types/domains are entirely unseen during training) and report performance on the held-out sets; additionally, include robustness tests and error analyses for cross-lingual cases and extreme layouts (e.g., scanned documents, multi-column pages, complex tables).
3. The data pipeline relies on GPT-5 for heuristic filtering and ternary comparisons (Synth-vs-Synth). Even with a small-scale human audit, this may distill upstream model preferences in layout/style into the training signal. Moreover, the paper does not disclose expert grading rubrics/annotation guidelines or quantify inter-annotator consistency.
4. The current “extrinsic evaluation” is primarily an offline candidate generation → reward-based reranking setup and does not demonstrate that the reward can serve as an effective training signal to substantively improve generation quality. It is recommended to add downstream tasks that optimize layout/page generation with this reward (e.g., as a signal for fine-tuning or RL), and report the resulting gains.

**Questions:**

1. Are the DOCPAIR dataset and its construction methodology open-sourced? If so, under what license?

---

> ### Author Response · Authors · 2025-11-25
> **Response to reviewer (1/4)**
>
> We appreciate reviewer Knxt's recognition of our clear problem specification, the scale and diversity of the DocPair dataset. We would like to clarify each of your points as follows.
>
> ---
>
> ### About Contributions
>
> We acknowledge that our work leverages existing multimodal backbone models, which is a common and necessary practice in the era of LLMs. However, our core contribution lies not in inventing a new model architecture, but in introducing crucial innovations in task definition, methodology, and evaluation paradigm. Specifically, our advances are as follows:
>
> 1. **Novel Task Formulation**. We formally define, for the first time, the evaluation of documents in terms of their visual **structural and stylistic professionalism,** a key yet underexplored problem. Unlike traditional assessments that focus on the professionalism of textual content  (e.g., grammar, coherence, factual accuracy), our work focuses on the structural and stylistic aspects of professionalism, which is critical for readability and engagement.
> 2. **Content-Agnostic Methodology**. We propose a novel data construction methodology that **decouples the quality of textual content from the professionalism of structure and style**. This is realized through the creation of paired documents where the textual content is identical, but the visual structure and style are varied. This decoupled data construction is a methodological innovation that makes it possible, for the first time, to purely evaluate a document's structure, and style, free from the interference of textual semantic quality.
> 3. **Advances in the Aesthetic Evaluation Paradigm**. Existing aesthetic assessment methods (e.g., AesthetiQ [1], LACE [2]) primarily target images, web pages, or UI interfaces, focusing on **single page** only. In contrast, our work is the first to focus on **multi-page documents**, where evaluation criteria encompass cross-page consistency (e.g., uniform headers/footers, margins, fonts) and complex structural organization (e.g., logical layout, formatting).
> 4. **Missing Piece in Professional Document Creation.** Although current LLMs/agents are good at generating high-quality textual content, they critically lack the necessary visual quality reward signal to optimize their output for structural and stylistic professionalism. This absence leads to generated documents that often fall short in terms of readability and engagement. DocReward is the first dedicated Reward Model designed to fill this critical gap.
>
> [1] Xueru Zhang, et.al. Aesthetiq: Enhancing graphic layout design via aesthetic-aware preference alignment of multi-modal large language models. In The Twelfth International Conference on Learning Representations (ICLR), 2024.
>
> [2] Yuxin Li, et.al. Towards aligned layout generation via diffusion model with aesthetic constraints. In The Eleventh International Conference on Learning Representations (ICLR), 2023.

---

> ### Author Response · Authors · 2025-11-25
> **Response to reviewer (2/4)**
>
> ### Out-of-domain Test and Cross-lingual Test
>
> 1. **Out-of-domain experiment.** According to the reviewer’s suggestion, we conducted additional out-of-domain experiments on unseen domains. The results are presented in the table below, from which we draw the following conclusions:
>
>    1. **In the out-of-domain setting, the DocReward-7B model achieves a high score of 85.55**, remaining superior to all other baseline models, including the closed-source model GPT-5 (71.11). This trend is consistent with the in-domain results.
>    2. **Superior Generalization of DocReward.** We observe that the DocReward model exhibits only a minimal performance drop when transitioning from in-domain to out-of-domain evaluations. The performance of DocReward-7B decreases by **merely 3.67** percentage points. Such a small performance gap indicates that DocReward generalizes effectively to entirely new and unseen domains.
>
>    |               | In-domain | Out-of-domain |
>    | ------------- | --------- | ------------- |
>    | Qwen2.5 VL-3B | 34.88     | 31.10         |
>    | Qwen2.5 VL-7B | 50.95     | 45.92         |
>    | GPT-4o        | 58.56     | 57.04         |
>    | GPT-5         | 69.77     | 71.11         |
>    | DocReward-3B  | 86.89     | 81.85         |
>    | DocReward-7B  | 89.22     | 85.55         |
>
>
>
> 2. **Cross-lingual experiment.** We conducted cross-lingual experiments to evaluate the performance of the DocReward model on non-English documents. The conclusions are as follow:
>
>    1. **Superior Performance of DocReward on Non-English Languages .** The DocReward-7B model still achieved a high score of 77.90, substantially outperforming all baseline models (exceeding GPT-5 (60.42) by 17.48 percentage points). This is consistent with the conclusions drawn from the English evaluation.
>    2. **Robustness in Cross-lingual Settings.** All models, including the baselines, exhibited performance degradation in non-English settings. For example, GPT-5 dropped by 9.35%. The performance drops of DocReward (−9.39%, −11.32%) are **comparable** to those of the closed-source models GPT-4o (−11.06%) and GPT-5 (−9.35%), indicating that DocReward demonstrates strong cross-lingual robustness.
>
> |               | French | Spanish | Rusian | Non-English Avg. | English | Drop    |
> | ------------- | ------ | ------- | ------ | ---------------- | ------- | ------- |
> | Qwen2.5 VL-3B | 35.00  | 33.75   | 22.50  | 30.42            | 34.88   | -4.46%  |
> | Qwen2.5 VL-7B | 48.75  | 42.50   | 27.50  | 39.58            | 50.95   | -11.37% |
> | GPT-4o        | 47.50  | 52.50   | 42.50  | 47.50            | 58.56   | -11.06% |
> | GPT-5         | 57.50  | 76.25   | 47.50  | 60.42            | 69.77   | -9.35%  |
> | DocReward-3B  | 77.50  | 82.50   | 72.50  | 77.50            | 86.89   | -9.39%  |
> | DocReward-7B  | 78.75  | 88.75   | 66.25  | 77.90            | 89.22   | -11.32% |

---

> ### Author Response · Authors · 2025-11-25
> **Response to reviewer (3/4)**
>
> ### Reliance on GPT-5
>
> 1. **GPT-5 is only used as human proxy via an triplet-based annotation protocol, instead of distillation.** When annotating training pairs, rather than directly querying GPT-5 for its preferences between two given documents, we adopt an **oracle setting** - namely, a triplet-based annotation protocol. In this setup, the original **human-authored document serves as a reference** against which two synthetic candidate documents are evaluated. Unlike employing GPT-5’s intrinsic preferences to decide which is better, this protocol assesses which candidate document more closely resembles the original high-quality one in terms of structure and style, which is objective and determined. We quantified the accuracy of the triplet-based annotation protocol with human-annotated comparison pairs and it achieves an **average accuracy of 92.5%**, so here GPT-5 is only used as human proxy instead of distillation.
> 2. **GPT-5 is only employ for train-set instead of test-set**. All samples in test-set are annotated by human annotators.
> 3. **Instead of distillation from GPT-5, DocReward achieves better performance than GPT-5 on human-annotated test-set.** On the human test-set, DocReward’s accuracy substantially surpasses that of closed-source LLM baselines (e.g., exceeding GPT-5 by 19.4 percentage points), as shown in Table 2. If DocReward were merely distilled from GPT-5, it **would not exhibit** **such a performance gain** over it.
>
> ---
>
> ### Annotation Guidelines, Inter-annotator Consistency
>
> 1. **Inter-annotator agreement.** As shown in table below, we evaluate annotation consistency among human annotators using the rigor metric of **Cohen's Kappa** and observe a high agreement of **83.40**. As described in [1], this is a quite high agreement. This high-level agreement suggests that the annotations are reliable.
>
>    |                 | Annotator 1 | Annotator 2 | Annotator 3 | Average |
>    | --------------- | ----------- | ----------- | ----------- | ------- |
>    | **Annotator 1** | -           | 83.40       | 80.92       | 82.15   |
>    | **Annotator 2** | 83.40       | -           | 85.90       | 84.65   |
>    | **Annotator 3** | 80.92       | 85.90       | -           | 83.41   |
>    | **Average**     | 82.15       | 84.65       | 83.41       | 83.40   |
>
>
> 2. The annotation guideline for human annotators are presented in Appendix A.4.
>
>
> [1] McHugh, Mary L. “Interrater reliability: the kappa statistic.” *Biochemia medica* vol. 22,3 (2012): 276-82.

---

> ### Author Response · Authors · 2025-11-25
> **Response to reviewer (4/4)**
>
> ### Reinforcement Learning Experiments of DocReward
>
> * As recommended by the reviewer, we further conduct reinforcement learning experiments to verify the effectiveness of DocReward. We perform **GRPO** [1] on an open-source model (i.e., Qwen-2.5-Coder) and recently proposed **Training-free GRPO** [2] on close-source model (i.e., GPT-4o).
>
> * Given plain, unformatted plain text as input, the model’s task is to generate a professional Word document by producing python-docx code [3] . The resulting document should have a professional structure and styling, such as appropriate headings, paragraphs, tables, font size and font type, etc.
>
> * The reward consists of two parts: the first measures the similarity between the generated document content and the original input text to check whether text content is lost (ROUGE-L); the second comes from the DocReward score, which encourages the model to produce documents that are **more professional, well-structured, and well-styled**. After RL, human annotators are asked to rank the documents produced by the six models in the table. The evaluation criterion is the professionalism of the document’s structure and styling. Other experimental details are presented in Section 4.4.
>
> * As shown in the following table, rule-based rewards substantially improve the success rate of document generation for both Qwen2.5-Coder and GPT-4o. Incorporating DocReward as an additional reward signal further enhances the performance of both open- and closed-source models, leading to **higher success rates, improved ROUGE-L scores, and better human rank**. These results demonstrate that DocReward serves as **an effective reward model** for strengthening document generation capabilities, especially in terms of professional structure and style.
>
> * | Method                                     | Success Rates↑ | ROUGE-L↑ | DocReward↑ | Human Rank↓ |
>   | ------------------------------------------ | -------------- | -------- | ---------- | ----------- |
>   | Qwen2.5-Coder                              | 30.0           | 20.61    | 0.0663     | 4.58        |
>   | - w/ GRPO (rule)                           | 98.0           | 97.94    | 0.1785     | 4.06        |
>   | - w/ GRPO (rule & DocReward)               | 100.0          | 97.95    | 0.3046     | 2.84        |
>   | GPT-4o                                     | 52.0           | 48.73    | 0.2682     | 3.18        |
>   | - w/ Training-free GRPO (rule)             | 66.0           | 62.15    | 0.3189     | 2.70        |
>   | - w/ Training-free GRPO (rule & DocReward) | 74.33          | 74.33    | 0.4486     | 2.02        |
>
> * For the visualization of documents generated by different models, please refer to Figure 5.
>
> ---
>
> ### About Open-sourcing
>
> Yes, both the DocPair dataset and the construction methodology for DocReward will be open-sourced under MIT license.
>
> ---
>
> [1] Zhihong Shao, et al. Deepseekmath: Pushing the limits of mathematical reasoning in open language models. arXiv preprint arXiv:2402.03300, 2024.
>
> [2] Yuzheng Cai, et al. Training-free group relative policy optimization. arXiv preprint arXiv:2510.08191, 2025.
>
> [3] https://python-docx.readthedocs.io/en/latest/

---

### Official Review · Reviewer_qN6s · 2025-11-03

**Soundness:** 3
**Presentation:** 3
**Contribution:** 2
**Rating:** 6
**Confidence:** 3

**Summary:**

The paper introduces DocReward, a multimodal reward model based on Qwen2.5-VL for evaluating document professionalism in structure (e.g., spacing, alignment) and style (e.g., fonts, headings), ignoring textual quality. It builds DocPair, a 117K paired dataset across 32 domains/267 types with identical content but varying visuals, using agentic expansion and GPT-5 ranking. Trained with Bradley-Terry loss on rendered images. Overall, advances agentic workflows for visually polished documents.

Strength:
- Rigorous multi-phase dataset pipeline with diverse domains provides a scalable benchmark for multimodal reward modeling.

Weaknesses:
- Heavy dependence on closed-source LLMs for agentic expansion and ranking risks propagating their stylistic biases, potentially undermining the model's independence.
- Human evaluation emphasis on win-rates overlooks potential over-optimization to annotator preferences, which may not generalize to broader cultural or accessibility standards.

Overall, the paper contributes a new dataset and a new model that may benefit the community. The method is not technically flawed per se. However, the paper lacks technical depth and novelty. Therefore, I recommend weak acceptance.

**Strengths:**

- Rigorous multi-phase dataset pipeline with diverse domains provides a scalable benchmark for multimodal reward modeling.

**Weaknesses:**

- Heavy dependence on closed-source LLMs for agentic expansion and ranking risks propagating their stylistic biases, potentially undermining the model's independence.
- Human evaluation emphasis on win-rates overlooks potential over-optimization to annotator preferences, which may not generalize to broader cultural or accessibility standards.

**Questions:**

N/A

---

> ### Author Response · Authors · 2025-11-25
>
> Thank you for the positive assessment of our work! We address each of your points below.
>
> ---
>
> ### Use of Closed-source LLMs
>
> * **Closed-source is used solely as a proxy for human annotation to enable scalable train-set construction.** When annotating training samples, rather than directly querying GPT-5 for its preferences between two given documents, we adopt an **oracle setting** - namely, a triplet-based annotation protocol. The prompt is presented in Appendix A.6 (triple-wise).  In this setup, the original high-quality **human-authored document serves as a reference** against which two synthetic candidate documents are evaluated. Unlike employing GPT-5’s intrinsic preferences to decide which is better, this protocol assesses which candidate document more closely resembles the original in terms of structure and style, which is objective and determined. To verify the effectiveness of this method, we quantified the alignment between the triplet-based annotation protocol and human-annotated pairs and it achieves an average accuracy of 92.5%, demonstrating strong alignment with human judgment.
>
> * **Different from train-set annotated by closed-source LLMs, all test-set samples are annotated by human** **annotators**.
> *  **Performance over closed-source LLMs instead of dependency.** On the test-set described above, DocReward’s accuracy substantially surpasses that of closed-source LLM baselines (e.g., exceeding GPT-5 by 19.4 points), as shown in Table 2. If DocReward were merely distilled from closed-source models, it **would not exhibit** such a performance gain over them.
>
> ---
>
> ### Generalization to Broader Cultural or Accessibility Standards
>
>
> 1. **The annotation guidelines are objective regardless of cultural background.** The annotation guidelines are general principles, which exhibits a high degree of cross-cultural applicability. For example, extremely narrow margins that result in an almost fully saturated page layout are widely regarded as unprofessional, regardless of cultural or regional context. The detailed annotation guidelines are presented in Appendix A.4.
>
> 2. **In dependence of different cultural and professional standards.** The annotation process are carried out by three Ph.D. students majoring in computer science, marketing, and mathematics. We calculate the inter-annotator agreement and the **Cohen's Kappa is 83.4**. As described in [1], this is a quite high agreement. The high-level agreement among annotators from different backgrounds demonstrates that the annotation relies on clear, well-defined rules that are **independent of the annotators’ professional or cultural backgrounds**, reflecting a high degree of generality and objectivity.
>
>    |                 | Annotator 1 | Annotator 2 | Annotator 3 | Average |
>    | --------------- | ----------- | ----------- | ----------- | ------- |
>    | **Annotator 1** | -           | 83.40       | 80.92       | 82.15   |
>    | **Annotator 2** | 83.40       | -           | 85.90       | 84.65   |
>    | **Annotator 3** | 80.92       | 85.90       | -           | 83.41   |
>    | **Average**     | 82.15       | 84.65       | 83.41       | 83.40   |
>
>
> [1] McHugh, Mary L. “Interrater reliability: the kappa statistic.” *Biochemia medica* vol. 22,3 (2012): 276-82.

---

### Author Response · Authors · 2025-12-03
**Rebuttal Summary**

Dear Area Chair,

We sincerely thank you for handling our submission and are grateful to all reviewers for their constructive feedback. During the discussion period, we carefully and thoroughly addressed every concern raised by the reviewers, providing detailed clarifications, additional experiments, and further analyses.

**Response to Reviewer qN6s (initial score: 6, no response):**
We explained in detail that the closed-source model served only as a proxy for human annotators, and that the annotation guidelines generalize well across different backgrounds.

**Response to Reviewer Knxt (initial score: 4, no response):**

1. We clarified our contributions from multiple perspectives, including the *Novel Task Formulation*, the *Content-Agnostic Methodology*, our *Advances to the Aesthetic Evaluation Paradigm*, and filling the missing piece in *Professional Document Creation*.
2. Following the reviewer’s suggestions, we conducted rigorous out-of-domain and cross-lingual experiments, demonstrating the generalizability and robustness of DocReward. As Reviewer UQBH noted, *“the cross-lingual experiments significantly strengthen the paper.”*
3. We clarified that GPT-5 was used solely as a proxy for human annotators.
4. We conducted a rigor inter-annotator agreement analysis and observed a high Cohen's Kappa of 83.4.
5. As recommended by the reviewer, we performed reinforcement learning experiments using DocReward to train document-generation agents, demonstrating DocReward’s effectiveness.

We believe our responses properly addressed all of Reviewer Knxt’s concerns, though the reviewer did not participate in the discussion before the early closure of the response period.

**Response to Reviewer UQBH (initial score: 4, increased to 6 after discussion):**
We provided detailed answers to the reviewer’s questions regarding example figures, prompts, and Word versions. The reviewer acknowledged that “most of my concerns have been addressed” and stated that *“the cross-lingual experiments significantly strengthen the paper”*, subsequently raising the score from 4 to 6.

**Response to Reviewer XQhC (initial score: 2, no response):**
The reviewer’s concerns focused on adding more baselines, annotator number and background, inter-annotator agreement, error analysis, and statistical significance. We provided detailed responses to each point, added more baselines, and conducted a rigorous error analysis, including out-of-domain and cross-lingual evaluations (as also highlighted by Reviewer UQBH, *"the cross-lingual experiments significantly strengthen the paper"*). We believe our answers properly address the reviewer’s questions and concerns, though the reviewer did not participate in the discussion before the early closure of the response period.

We deeply appreciate the reviewers’ constructive feedback, which helped to significantly strengthen the paper. We sincerely hope you will take these clarifications and additional results into consideration. If you have any further questions, we would be more than happy to provide additional explanations.

Best,

Authors

---

### Note · Authors · 2026-01-05

I have read and agree with the venue's withdrawal policy on behalf of myself and my co-authors.